# Disentangling Domestication from Food Production Systems in the Neotropics

Charles R. Clement [1,*], Alejandro Casas [2,*], Fabiola Alexandra Parra-Rondinel [3], Carolina Levis [4], Nivaldo Peroni [4,5], Natalia Hanazaki [4,5], Laura Cortés-Zárraga [6], Selene Rangel-Landa [2], Rubana Palhares Alves [7], Maria Julia Ferreira [8], Mariana Franco Cassino [9], Sara Deambrozi Coelho [10], Aldo Cruz-Soriano [11], Marggiori Pancorbo-Olivera [12], José Blancas [13], Andrea Martínez-Ballesté [6], Gustavo Lemes [14], Elisa Lotero-Velásquez [6], Vinicius Mutti Bertin [15] and Guilherme Gerhardt Mazzochini [16]





1. Instituto Nacional de Pesquisas da Amazônia, Av. André Araújo, 2936—Petrópolis, Manaus 69067-375, Brazil
2. Instituto de Investigaciones en Ecosistemas y Sustentabilidad, Universidad Nacional Autónoma de México (Campus Morelia), Antigua Carretera a Pátzcuaro No. 8701, Col. Ex Hacienda de San José de la Huerta, Morelia 58190, Mexico; srangel@cieco.unam.mx
3. Departamento Académico de Biología, Facultad de Ciencias, Universidad Nacional Agraria La Molina, Av. La Molina, s/n—La Molina, Lima 15024, Peru; fabiolaparra@lamolina.edu.pe
4. Programa de Pós-Graduação em Ecologia, Departamento de Ecologia e Zoologia, Universidade Federal de Santa Catarina, Campus Universitário, s/n—Trindade, Florianópolis 88040-970, Brazil; carollevis@gmail.com (C.L.); nivaldo.peroni@ufsc.br (N.P.); natalia.hanazaki@ufsc.br (N.H.)
5. Departamento de Ecologia e Zoologia, Universidade Federal de Santa Catarina, Campus Universitário, s/n—Trindade, Florianópolis 88040-970, Brazil
6. Jardín Botánico, Instituto de Biología, Universidad Nacional Autónoma de México, Circuito Exterior, Ciudad Universitaria, Coyoacán 04510, Mexico; zarraga@ib.unam.mx (L.C.-Z.); andrea.martinez@ib.unam.mx (A.M.-B.); elisa.lotero@st.ib.unam.mx (E.L.-V.)
7. Programa de Pós-Graduação em Ecologia, Instituto Nacional de Pesquisas da Amazônia, Av. André Araújo, 2936—Petrópolis, Manaus 69067-375, Brazil; rubanapalhares@gmail.com
8. Programa de Pós-Graduação em Etnobiologia e Conservação da Natureza, Universidade Federal Rural de Pernambuco, Rua Dom Manoel de Medeiros, s/n—Dois Irmãos, Recife 52171-900, Brazil; ferreira.julia2208@gmail.com
9. Programa de Pós-Graduação em Botânica, Instituto Nacional de Pesquisas da Amazônia, Av. André Araújo, 2936—Petrópolis, Manaus 69067-375, Brazil; marianafcassino@gmail.com
10. Rua Alegria, 72—Centro, Aracruz 29190-230, Brazil; saradeambrozi@gmail.com
11. Coordinadora de Ciencia y Tecnología en los Andes—CCTA, Camilo Carrillo 300-A, Lima 15072, Peru; aldojcruz@gmail.com
12. Centro de Investigaciones de Zonas Áridas, Universidad Nacional Agraria La Molina, Jr. Camilo Carrillo 300-A—Jesús María, Lima 15072, Peru; 20120121@lamolina.edu.pe
13. Centro de Investigación en Biodiversidad y Conservación, Universidad Autónoma del Estado de Morelos, Av. Universidad 1001, Colonia Chamilpa, Cuernavaca 62290, Mexico; jose.blancas@uaem.mx
14. Curso de Graduação em Ciências Biológicas, Universidade Federal de Santa Catarina, Campus Universitário, s/n—Trindade, Florianópolis 88040-970, Brazil; gulemes50@gmail.com
15. Programa de Pós-Graduação em Ciências de Florestas Tropicais, Instituto Nacional de Pesquisas da Amazônia, Av. André Araújo, 2936—Petrópolis, Manaus 69067-375, Brazil; viniciusmbertin@gmail.com
16. Departamento de Biologia Vegetal, Instituto de Biologia, Universidade Estadual de Campinas, Rua Charles Darwin, s/n—Cidade Universitária, Campinas 13083-863, Brazil; gmazzochini@gmail.com
* Correspondence: cclement@inpa.gov.br (C.R.C.); acasas@cieco.unam.mx (A.C.)

**Abstract:** The Neolithic Revolution narrative associates early-mid Holocene domestications with the development of agriculture that fueled the rise of late Holocene civilizations. This narrative continues to be influential, even though it has been deconstructed by archaeologists and geneticists in its homeland. To further disentangle domestication from reliance on food production systems, such as agriculture, we revisit definitions of domestication and food production systems, review the late Pleistocene–early Holocene archaeobotanical record, and quantify the use, management and domestication of Neotropical plants to provide insights about the past. Neotropical plant domestication relies on common human behaviors (selection, accumulation and caring) within agroecological systems that focus on individual plants, rather than populations—as is typical of agriculture. The early archaeobotanical record includes numerous perennial and annual species,

many of which later became domesticated. Some of this evidence identifies dispersal with probable cultivation, suggesting incipient domestication by 10,000 years ago. Since the Pleistocene, more than 6500, 1206 and 6261 native plant species have been used in Mesoamerica, the Central Andes and lowland South America, respectively. At least 1555, 428 and 742 are managed outside and inside food production systems, and at least 1148, 428 and 600 are cultivated, respectively, suggesting at least incipient domestication. Full native domesticates are more numerous in Mesoamerica (251) than the Andes (124) and the lowlands (45). This synthesis reveals that domestication is more common in the Neotropics than previously recognized and started much earlier than reliance on food production systems. Hundreds of ethnic groups had, and some still have, alternative strategies that do involve domestication, although they do not rely principally on food production systems, such as agriculture.

**Keywords:** Amazonia; Andes; cultural niche construction; ethnobotany; ethnoecology; human selection; landscape domestication; Mesoamerica; plant domestication; plant management

---

## 1. Introduction

　　Humans are the dominant animal on the planet [1]. This simple affirmation has generated an enormous body of theory and research to explain why. One popular theory is that in the early Holocene humans domesticated plants and animals to create agricultural systems that fueled population growth, which led to social hierarchy, urban development and the states that appeared in the middle Holocene [2–4]. A theory described in a single sentence, with a linear sequence of events. Is it true? After all, to create a food production system, such as agriculture, one must first have good domesticates [5], i.e., domesticates that yield well in response to human labor. An ever-expanding body of evidence has shown that domestication is not linear, nor quick [6–8], and that many human societies did not produce the majority of their food, they managed ecosystems and collected what they wanted [9]. Cultivation, the basis of food production systems, has been called a slow evolutionary entanglement of humans and some of their plants [10,11]. The earliest cultivation systems most probably emerged from ancient forms of ecosystem management [2,12,13], where the eventual domesticates and other non-domesticated companions were progressively modified by both natural and human selection and other evolutionary forces [14–17]. Not all forms of ecosystem management became cultivation systems and not all resources managed in those ecosystems became domesticates, as documented by current ethnobotanical and evolutionary ecological studies [18]. Archaeological and ethnobiological studies have documented ancient and current processes of domestication [19,20], as well as forms of cultivation of plants in forested ecosystems [18,21]. Some of these processes and systems have remained as incipient domestication and as silvicultural systems, respectively, for centuries or millennia. Given these observations that do not agree with the popular narrative of the rise of states, we will incorporate more ethnobotany and ethnoecology to continue to disentangle domestication from reliance on food production systems in the Neotropics in order to contribute to the ongoing deconstruction of this standard narrative.

　　Archaeologists found that numerous late Pleistocene sites in southwest Asia provide evidence that people were managing and even cultivating populations of plants that did not show evidence of domestication. This was called pre-domestication cultivation [22]. The missing evidence is any morphological trait of the domestication syndrome that proves that change has occurred. In most definitions of domestication (Appendix A), people select—consciously and unconsciously—for a small number of traits that make their selected plants different from wild ones [23]; this set of traits is the domestication syndrome [24,25], which varies in composition among the different species that humans manage and cultivate. The term pre-domestication cultivation essentially assumes that the domestication process had not started because there were no changes observed in the plants in the archaeological record.

Geneticists examined the biological model that underlies change in any trait and inferred that selection intensity was extremely low at the beginning of the process [7]. They also found that selection must have started in the late Pleistocene, rather than the Holocene, in order to explain the observed changes in the domestication syndrome in the early-mid Holocene [6,7]. Crop genomes also suggest that the process was very slow, contrary to expectations based on cultivation, so other forms of population management must have been involved [26]. These studies confirm that there is evidence, even though it is invisible in the archaeological record. Other crop genomes have shown that introgression, hybridization among populations of the same species, is common, proving that the process is non-linear [27]. They also show that bottlenecks, the dramatic reduction in genetic variability attributed to the selection of a small number of progenitors to start cultivating, are probably the exception, instead of the rule [28–30]. All this research is also showing that plant domestication is more complex than commonly believed.

This raises the question: what is domestication? Anthropologist David Rindos proposed that domestication is an example of coevolution in which humans are involved. "Coevolution is an evolutionary process in which the establishment of a symbiotic relationship between organisms, increasing the fitness of all involved, brings about changes in the traits of the organisms", ([5] p. 99). Domestication is the process that creates a mutualistic relationship among a human population and populations of plants, where human selection and management change traits of interest to humans, such as fruit size, color, and sweetness. The genetic models mentioned above predict that selection intensity was very small at the beginning of the process, so another question is: when is a population domesticated? Archaeologists and geneticists rely on measurable changes in the domestication syndrome [19]. However, as pointed out by Rindos and shown by the models, small selection intensities result in small changes in traits. Initially, these changes are within the range of variation of the traits in wild populations ([2] p. 64), but are essential for continued change. Current studies of on-going processes of domestication have documented that gene flow between wild and managed populations may counterbalance and even mask the results of human selection. Therefore, it is not only a question of the selection intensity but also the intensity of other evolutionary and ecological forces operating in the system [14,15,31].

Domestication should not be analyzed from a purely anthropocentric perspective, since it is a co-evolutionary interaction between social and natural systems, and it influences the structure and dynamics of both. What has not yet been well explored is the agency of the plants and animals that became entangled with humans, as they have needs that humans must satisfy if the humans hope to benefit from their relationships with these non-humans [32–34]. Many, if not most, Neotropical ontologies recognize that non-humans have agency [35–37]. In Neotropical ontologies, however, non-human agency is more than mere adaptation to culturally constructed niches; non-humans are active subjects of landscape transformations [38,39], and humans must negotiate with them for these entanglements to satisfy all members of the niche [37]. These observations are not considered in the standard narrative.

As elsewhere, the Neotropics never had a Neolithic Revolution [37,40,41], although the standard narrative influences research and interpretation also. The region saw the rise and fall of states, as well as the domestication of numerous native crops and a few animals. All the states arose in the Neotropics, although other parts of the Americas were home to a diversity of societies also. Our focus is in three parts of the Neotropics: Mesoamerica, home to the Olmec, Teotihuacan, Maya, Mixtec-Zapotec, Aztec and numerous other societies; the Andes, home to the Caral, Wari, Chavin, Chachapoyas, Nazca, Tiwanaku, Inca and numerous other societies; and the South American lowlands, home to numerous societies that did not become states [42].

Mesoamerica is a term proposed by Paul Kirchoff [43] to designate the cultural area influenced by the pre-Columbian states that developed in central and southern Mexico and northern Central America. The area was dynamic through time, and by the arrival of

the European conquerors it included the area from southern Mexico, with two northern fringes along the coasts of the Pacific Ocean and the Gulf of Mexico, to north-western Costa Rica [44]. We include other parts of northern Mexico and southern Central America in our focus. The Andes does not have a similar cultural designation, although it was similarly dynamic culturally, especially along the Pacific coast and later in the Altiplano of Bolivia and Peru. The eastern lowlands include the Amazon forests and savannas, the Brazilian savanna and grasslands, semi-arid forests, and the Atlantic forest. In Amazonia, several expansive language families originated and some societies created monumental earthworks, such as geoglyphs [45]. Note that Kirchoff's cultural area and similar phenomena in the Andes and the eastern lowlands are all relatively late in pre-history. They are phenomena of what is called the Formative in American archaeology. The Formative started 4000 to 3000 years before present (y BP) and is identified by the increasing reliance on food production systems by some societies [40]. Before the Formative, a variety of societies practiced "low-level food production" [46], a concept with some similarity to "pre-domestication cultivation", and part of the deconstruction of the standard narrative. We will not examine the Formative archaeobotanical record; instead, we will expand Piperno and Pearsall's [40] review of the earliest records of crops during the late Pleistocene and early Holocene to continue to disentangle domestication from reliance on food production systems.

Soviet geneticist and biogeographer Nicolay I. Vavilov considered Mesoamerica and the Andes to be centers of crop genetic diversity, where numerous crops were domesticated and even more were accumulated [47]. The Soviet expeditions pioneered what we call ethnobotanical crop surveys today, i.e., they inventoried plant knowledge and use in numerous localities throughout Mesoamerica and the Andes, identifying dozens of crops that might be useful in the Soviet Union and hundreds of crops and other useful plants found in each locality. The Soviet expeditions did not visit the eastern lowlands, because they considered that the great majority of crops and other useful species would not adapt in Soviet ecosystems, but Vavilov recognized that the area contained a wealth of crops and other useful plants [48]. Later, several centers of crop genetic diversity were proposed in Amazonia [49,50]; these are centers of accumulation, and seldom coincide with the origins of Amazonian crops [51]. This lack of coincidence between centers of origin and centers of accumulation makes the eastern lowlands quite different from Mesoamerica and the Andes, stimulating Harlan to consider South America to be a non-center [52]. Following Vavilov, we will focus on the wealth of useful plants in the Neotropics, including species with populations domesticated to various degrees, species with managed populations, and the thousands of other useful species in these regions.

After the studies by Vavilov, numerous studies documented the antiquity of early signs of domestication of numerous crops, evolutionary relationships between crops and wild relatives, biogeographic distribution of wild relatives, the areas where domestication and diversification of crops were particularly dynamic, different types of landscape management, and early intricate relationships and interchange of experiences of managing ecosystems and plant populations among areas of the Neotropics [40,53–55]. All these studies suggest that a broad spectrum of domestication of plants and landscapes was involved in food procurement and production in the Neotropics. These studies also highlighted that ethnobotany and evolutionary ecology can contribute at different scales: (1) through time, since the present serves as a baseline to look into the past; (2) across local space, since landscapes and ecological communities interact with human–plant entanglements; and (3) across the Neotropics, since humans can be extremely mobile and take their non-human entanglements with them.

Our contribution includes: (1) an examination of the process of domestication to identify the human behaviors that drive it; (2) a review of the Neotropical archaeobotanical literature to identify the earliest human–plant interactions that would lead to domestication of plant populations of these species; and (3) a preliminary quantification of Neotropical plant use, management and domestication in areas where states arose and where other societies existed without relying on food production systems.

## 2. Definitions of Concepts

As pointed out by Piperno and Pearsall ([40] p. 6), clear definitions of concepts are extremely important when discussing domestication and food production systems. A short compilation of definitions of domestication highlight recent variability in detail and terminology (Appendix A). Melinda Zeder [56] reviewed an earlier set of definitions to highlight their differences and theoretical frameworks. Even our grammar influences the way we interpret simple phrases about domestication ([34] p. xiv), as we tend to put ourselves, as individuals or the human collective, in charge. Recent concepts and full definitions can help identify nuances of our definitions that place human culture within Nature, as occurs in most Neotropical ontologies [57]. We think that returning to a dictionary is also an appropriate exercise to explore definitions, especially as the definitions in Zeder [56] and Appendix A were designed by their authors to meet their own objectives, the majority of which are associated with the domestication syndrome.

Darwin used domestication as a metaphor for natural selection because everyone is familiar with the term. The verb domesticate comes from the Latin *domesticäre*—to dwell in a house, to accustom. The house is the center of the *domus*, the Latin root of *domesticäre*. The Oxford English Dictionary (OED) definitions include: "1.a. To make, or settle as, a member of a household; to cause to be at home; to naturalize. 1.b. To make to be or to feel 'at home'; to familiarize. 2. To make domestic; to attach to home and its duties. 3. To accustom (an animal) to live under the care and near the habitations of man; to tame or bring under control; to civilize. 4. To live familiarly or at home (with); to take up one's abode". It follows that the noun domestication is "the action of domesticating, or the condition of being domesticated" (OED), i.e., both the process itself and the results of the process.

These definitions are about humans [58], who domesticate each other in their houses, with their associated gardens, orchards, pastures, woodlots, agroforests, and adjacent managed forests. As such, the house and the surrounding landscape comprise the *domus* [3]. Animals are recognized in one of the definitions as being domesticated in the *domus* as well, and it was Darwin who included plants. From these definitions, domestication is clearly anthropocentric.

What are the human behaviors involved? Two are explicit in the definitions: care (of the occupants of the *domus*) and duty (the tasks of caring for the *domus* and its occupants). Two are implicit. Selection—since humans are selective about what is brought into the *domus*. Accumulation—as people and animals are brought into the *domus*, both from nearby (familiarize) and far away (naturalize). The definitions are about both organisms (humans, animals, plants) and the *domus*. Most current definitions of domestication (Appendix A) only mention one behavior (selection), but care and accumulation are just as important [59].

### 2.1. Domestication as Process

Rindos theorized that these common behaviors are sufficient to start the domestication process that interested Darwin, within which Rindos defined three stages. "In the simplest terms, incidental domestication includes simple dispersal and protection actions by people that create and maintain coevolutionary interactions outside the agroecology; specialized domestication focuses on the forces initiating and maintaining the agroecology; and agricultural domestication is largely concerned with the forces controlling the function, evolution, and spread of developed agricultural systems ( . . . )." ([5] p. 153). Observe that selection and caring, in the form of protection, are at the beginning of the process. For Rindos, an agroecology originates where humans create conditions for the growth of useful plants, which is conceptually similar to Lewontin's ideas of niche construction ([5] pp. 142–143); Piperno and Pearsall ([40] p. 6) call this cultivation. This is also caring or nurturing [59]. The creation of an agroecology starts in a new settlement, especially with its associated dump heaps, and gradually extends outwards. As Rindos points out, this does not have to be done intentionally for the benefit of the plants, but it creates new ecological conditions, or niches, and some plants will take advantage of them. These colonizing species, sometimes called weeds, benefit from the processes that create agroecologies and some of these

species are useful to humans, so they are maintained and protected [14,15,60–62]; they may be selected and become domesticated [2,32,62].

Rindos hypothesized that the process started during the Pleistocene, which for our purposes is the arrival of humans in the Neotropics. Upon arrival, humans identified useful species and the variation within each, which allows the preliminary selection of individuals for gathering. The gathering starts dispersal towards the settlement ([5] pp. 112–120) [63,64], as some fruit or seeds may fall en route, or may be consumed and the seeds excreted en route [65–67]. The best individuals in the population (and later those dispersed en route) are protected, others may be tolerated, the worst may be eliminated if they compete with preferred individuals or good individuals of other useful species [67]. All of these activities gradually result in a population more useful to humans in the original ecosystem, which is gradually becoming a domesticated landscape—all without creating an agroecology, hence Rindos' term incidental domestication ([5] pp. 154–158).

As mentioned, the agroecology, with its specialized domestication ([5] pp. 158–164), appears in human settlements, all of which have dump heaps. The dump heap origin of crop domestication and food production has a long history [68], which we will not review. It is sufficient to recognize that dump heaps and nearby areas of the settlement develop into gardens [59,68,69], which are recognizable agroecologies that can be replicated beyond the settlement, giving rise to more intensive agroecologies where agricultural domestication continues ([5] pp. 164–166). The latter is what most people consider to be domestication, as seen in the standard narrative of crop domestication and agricultural origins (see also Appendix A). By recognizing only the latter stage, a long history of human–plant interactions is ignored.

The concept of domesticated landscapes has been used for a century with various names (see synonyms in Smith [17]), and can conceptually be disentangled from domestication of plants and animals [70]. The concept can be defined as a process by which human manipulation of the demography of plant and animal populations changes the landscape's ecology, resulting in a landscape more productive and congenial for humans [70,71]. At its simplest, the protection and dispersal typical of incidental domestication promote the initial changes ([5] pp. 112–120) [17,64]. As management intensity increases, with the removal of competitors, intentional planting of seeds and seedlings, mulching around these, and other practices to care for individuals, the landscape becomes more productive [67]. Notice that a continuum of change is evident, as humans invest more effort in caring for some useful plants. These changes are important for both humans and plants, and create conditions favorable for other species of plants and animals as well [72,73].

A crucial addition in the process may be cultivation, which introduces more dramatic changes in the ecosystem, as noted by Rindos. Although cutting trees comes to mind first, William Denevan [74] pointed out that with a stone axe it is easier to find a clearing where a large tree fell or a wind storm had opened a larger area. Once open, fire becomes an essential tool ([3] pp. 37–43) [12,17,74]. All the practices already mentioned are used to propagate and care for useful plants in a new agroecology. Today this generally starts as a horticultural plot and turns into an agroforestry plot, often seen as mimicking natural ecosystems, especially as local species volunteer and are tolerated in the agroecology [75].

"Since domestication is an evolutionary process, there will be found all degrees of plant and animal associations with man and a range of morphological differentiations from forms identical to wild races to fully domesticated races. A fully domesticated plant or animal is completely dependent upon man for survival." ([2] p. 62). This observation calls for a definition of the domestication continuum, rather than using a general definition that does not discriminate any possibilities along the way except the last one (many definitions in Appendix A), when the domestication syndrome is clearly visible. One such definition, following from Rindos, is that plant domestication is a coevolutionary process during which human selection of the phenotypes of wild, promoted, managed or cultivated plants results in changes in the next generation's phenotypes and genotypes that make them more useful to humans and better adapted to domesticated landscapes [71]. Notice that Harlan

affirmed that during the beginning of the process the changes are so subtle that they are hard to differentiate from wild populations ([2] p. 64), with the clear implication that the domestication syndrome contains only incipient changes in one or a few traits, and may only be visible as reduced variability [71].

*2.2. Domestication as Result (the Domestication Syndrome)*

　　The results of the domestication continuum extend from the first incipient changes to a clearly differentiated domestication syndrome [71], and are the major interest of the definitions of most authors presented in Appendix A. Along the continuum, some categories can be identified (Figure 1), mostly for convenience in discussing the concept. An incipiently domesticated population has both the mean and variance of a selected trait within the variation of the species ([2] p. 64). A semi-domesticated population has more pronounced differentiation, while a domesticated population may extrapolate the variation of the wild populations and also has become dependent on humans (the last stage mentioned by Harlan). Although this sequence is defined as being linear, the world is much more complex. There are multiple ways to get from A to D, and there is no guarantee that C or D will be attained or, if attained, maintained through time.

　　During the mid-20th century most students of domestication thought that the process could be quite rapid, e.g., from wild to domesticated phenotypes, such as the non-shattering seed rachis of wheat (*Triticum* spp.) or barley (*Hordeum vulgare*), in 200 years [2]. This fitted nicely with the standard narrative about the rise of civilizations. Since the turn of the millennium, archaeologists found that the wheat and barley domestication processes took thousands of years [3,10,76], and that "pre-domestication cultivation" occurred in south-western Asia in the late Pleistocene [22], similar to the situation for "low-level food production" in the Americas [46]. At the same time, geneticists used the biological model of domestication to identify how human behaviors interact with plant genetics. The biological model has two interacting equations [77]:

$$V_P = V_G + V_E + V_{GxE} \tag{1}$$

$$R = h^2 \times i \times \sqrt{V_P} \tag{2}$$

　　Equation (1) explains the relationship among variances of phenotypes ($V_P$), genotypes ($V_G$), environment ($V_E$) and the genotype-by-environment interaction ($V_{GxE}$); this is the variability that fascinated Darwin [23] and the humans who accumulate it. As in Figure 1, these variances are of any trait of interest to humans in a population, e.g., fruit size. This equation is about what is available to humans ($V_P$) and explains how phenotypes are created during growth depending upon genotypes and their interactions with their environment. Equation (2) explains the response (R) when humans select (i) from the population. The narrow sense heritability ($h^2$) is that proportion of $V_G$ that explains the similarity between parents and offspring for the trait of interest [77]. The greater $h^2$, i, $V_P$, the greater the response; reduce any variable and the response decreases. Each trait of the domestication syndrome can be analyzed this way.

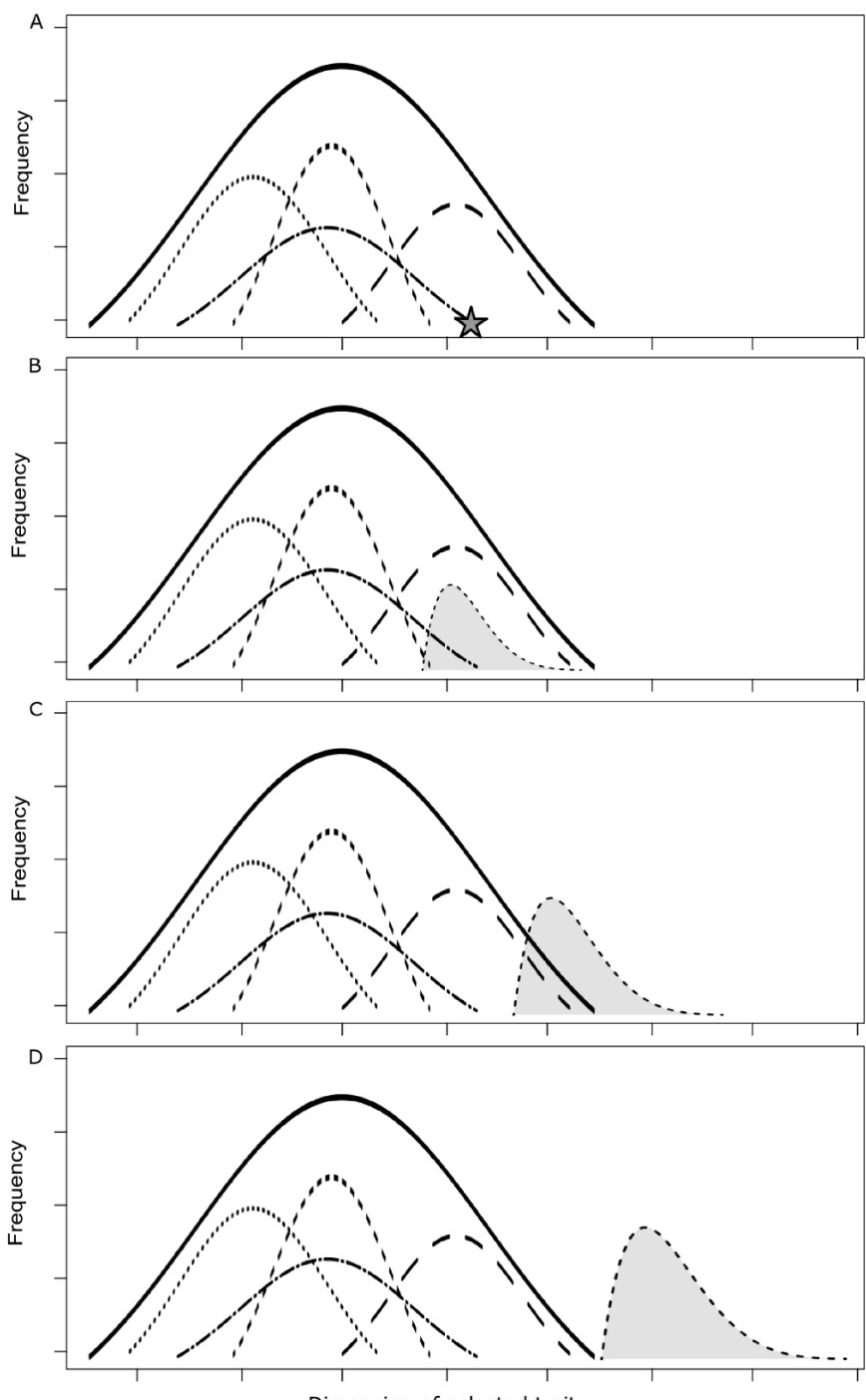

**Figure 1.** Hypothetical domestication continuum (frequency distributions for population means and variances of the dimension of a phenotypic trait of interest to humans, e.g., fruit size). (**A**) A species with four wild populations. A few plants are selected in one population (star) to create a new population in (**B**). (**B**) As above, with one incipiently domesticated population. Observe that the variance of the incipient domesticate is smaller than that of the wild populations, due to the selection of a small number of plants to create the new population—a result of the founder event. Observe also that the mean and variance are within the variance of the species. (**C**) As above, with one semi-domesticated population with somewhat greater variance. (**D**) As above, with one domesticated population. Observe that both the mean and the variance are outside of the variance of the species. Observe in (**B**–**D**) that the domesticated populations have skewed distributions, with more variation towards the right side, representing directional selection, e.g., for larger fruit. Adapted from Leakey et al. [78], with thanks to Alessandro Alves-Pereira.

The model used by geneticists found that the intensity of selection for the non-shattering rachis of wheat and barley was extremely low, only slightly different from that expected from natural selection, and that it took thousands of years for the non-shattering trait to reach even low proportions in the pre-domestication cultivated populations [6–8]. These authors did not use the term, but this is what we mean by incipient domestication, a change so small that it is hard for archaeologists to identify.

Most geneticists and many archaeologists who study domestication do not give much attention to an extremely important component of the model: $V_E$. Rindos [5], however, was very clear that plant domestication occurs within domesticated landscapes and that, as the human investment in agroecological management expands, so does the response to selection. Equation (1) states that $V_P$ is the sum of three other variances and can be modified by changes in any of them. Like $V_G$, which contains several genetic components, $V_E$ contains numerous biotic and abiotic components typical of niches [79,80], such as soils, water availability, pollinators, pests, diseases, herbivores. Humans and their management practices are also biotic and social components of the niche, which becomes an agroecological niche as human action increases. Choices about where to plant, when to plant, how to fertilize, irrigate, weed, etc., all affect $V_E$ and feed into $V_P$, both directly and via $V_{GxE}$ [81]. Since these can change $V_P$ without human selection (i), it is possible to obtain a response (R) without human selection, so management of the agroecology is always an important consideration. Observe also that all these management options are designed to meet the needs of plants who respond to this care, e.g., this response is plant agency [34]. When there is human selection, management practices enhance the response.

### 2.3. Food Production Systems

There are numerous types of food production systems [82], each with somewhat different agroecologies and human decisions about crops and their management. The two most commonly used terms are horticulture and agriculture. Horticulture comes from the Latin *hortus* (garden) and *cultūra* (culture or cultivation), and is defined as "The cultivation of a garden; the art or science of cultivating or managing gardens, including the growing of flowers, fruits, and vegetables" (OED). Fruits are often produced by trees, which have their own term: arboriculture (from the Latin *arbor*—tree). Since many fruits and other products are also produced on plants that are not trees (e.g., palms, cacti, agaves) and which occur in some types of forest, it is also appropriate to define silviculture (from the Latin *silva*—forests or stands of trees). Similarly, agriculture comes from *agrī* (genitive of *ager*—field) and is defined as "(a) Originally: the theory or practice of cultivating the soil to produce crops; an instance of this (now rare). (b) Later also (now chiefly): the practice of growing crops, rearing livestock, and producing animal products (as milk and eggs), regarded as a single sphere of activity; farming, husbandry; (also) the theory of this", (OED). In modern usage, agriculture is thus all inclusive, but its original use was for crops, which are defined as "The annual produce of plants cultivated or preserved for food, esp. that of the cereals; the produce of the land, either while growing or when gathered; harvest", (OED). This is why agriculture is generally associated with southwestern Asia, where wheat and barley were domesticated in fields, although gardens, including with cereals ([3] p. 43), other annuals and perennials, were certainly earlier than fields, although seldom emphasized (e.g., [2,83]).

Agriculture is the term of choice in the standard narrative about the rise of states. In this narrative, horticulture is small-scale (gardens), even "primitive", compared to agriculture, which is large-scale (fields), with advanced technologies, such as draft animals to pull plows and operate threshing equipment to separate chaff from grain, etc. Scale often decides the usage [82,84]. A recent article about the expansion of maize (*Zea mays*) use and production in pre-Columbian Mexico asks "Is it agriculture yet?" [85], referring directly to the scale of use and production. A majority of scholars follow this usage, e.g., [40].

Since we are discussing domestication, however, another factor is important. Oake Ames [86], cited by Rindos [5] and Leach [82], observed that in horticulture plants are

treated as individuals, while in agriculture plants are treated as groups (or populations). This is an extremely important observation because it has to do with selection (i) and thus response to selection (R) in the biological model. Those of us with gardens often talk or sing to our plants and they respond, especially if we weed, fertilize and irrigate as we talk. Anthropologists have reported this among indigenous peoples across the Neotropics [82,87,88], where local ontologies consider other living beings to be social organisms similar to humans in many respects [35]. In Amazonia, indigenous women consider their manioc (*Manihot esculenta*) plants to be their children and sing to them to encourage them [89]; in Mexico, the Mixtec, Nahua and other peoples pray in their milpas to encourage and safeguard the maize while they care for the plants [90]. This caring also implies a duty to care for and protect. Although there is an effect on $V_E$, due to the weeding, fertilizing and irrigating, what is more important is selection (i), because the better you know your individual plants the easier it is to decide which ones get more space in the next garden. This is true for plants propagated by seeds (e.g., maize) or vegetatively (e.g., potatoes (*Solanum tuberosum*) or manioc). The cultivation of manioc and potatoes is a special kind of horticulture, called vegeculture—the culture of vegetatively propagated plants [91,92]. Harlan ([2] p. 131) observed that vegetative propagation is instant domestication, because the plants depend completely on their humans to be propagated into the next garden.

Another factor is important in the Neotropics: no animals were domesticated that could pull a plow. As such, all labor in food production systems was human, although ducks may have helped with weeding and pest control [93]. Neotropical societies had numerous tools for working the soil and processing plants, but none that permitted the scale typical of agriculture with draft animals. That is not to say that they produced less food; the early chroniclers marveled at the well-feed, healthy people in the villages and urban areas they conquered [94,95]. In some places, there were moderately large individual fields, such as the raised fields in the Llanos de Mojos, lowland Bolivia, which could be $20 \times 50$ m, with dozens of such raised fields around some villages [12]. In Central West Mojos, an area of about 10,000 km$^2$, there are about 36,000 raised field platforms, with a total raised-surface area of 100 km$^2$ ([96] p. 105). In the Andes, the tens of thousands of terraces, *andenes*, each had small surfaces, many even smaller than in Mojos, but summed were able to support the Inca state [12]. Similarly, the milpa and agroforestry systems of the Maya supported its large population [97,98], and the terraces, chinampas and agroforestry systems of the Aztecs supported another large state [99,100]. In Amazonia, the *chacra* horticultural plots and agroforestry systems supported the expansion of the Arawak-speaking peoples [101], and were used by all other ethnic groups that decided to practice horticulture and agroforestry. Importantly, in the more forested Neotropical regions, including the Atlantic Forest and forested savannas in Brazil, these agroforestry systems were complemented by forest management [21,67,98,102], and some, perhaps many, societies obtained more food from their forests than from their gardens and agroforests [37,103].

The observant reader will have noticed that an additional term slipped in: agroforestry. Unlike the other terms we have used, this is not derived directly from Latin, but from research on modern small-scale indigenous and traditional food production systems across the tropics [104]. The term suggests a combination of agriculture with forestry. The majority of the hundreds of different agroforestry systems described by PK Nair [104] are in reality combinations of horticulture, vegeculture, arboriculture and silviculture, generally with volunteer plants that are tolerated and may be protected.

In this study we use the term food production system, even for the large-scale systems that supported the rise of Neotropical states. This decision is based on how Neotropical people treat their plants, but is also political, because we hope to continue to disentangle domestication from the standard narrative about the rise of states.

### 3. Early Human Entanglements in the Neotropics

The Americas were the last continents occupied by modern humans in our expansion across the planet. Recent research in an archaeological site in north-central Mexico, Chiquihuite Cave, supports previous suggestions that arrival was shortly before the last glacial maximum (LGM), perhaps as early as 33–30 k BP [105]. This site yielded a large number of stone artifacts, some charcoal, plant and animal DNA, plant phytoliths and pollen. Some of the phytoliths are attributed to the palm *Brahea berlandieri* and are burned, suggesting use as food (it has edible fruit [106]; see also Rangel-Landa et al. [107] for *Brahea dulcis*).

According to genetic data, human populations called southern Native Americans (SNA) occupied western North America and migrated into Central and South America during the late Pleistocene (inferred between ~17 and ~13 k BP) [108] (Figure 2). When SNA arrived in Panama, some groups entered South America along the Pacific Ocean coast and other groups along the Atlantic Ocean coast, from where each penetrated the interior. The Pacific lineage occupied the coast and the Andes, and reached the famous Chilean archaeological site of Monte Verde by ~18.5 k BP [109]. The Atlantic lineage occupied the eastern lowlands, and reached the central Brazilian archaeological site of Santa Elina between ~27 and ~23 k BP [110,111]. Mixture between the Andean populations and the eastern lowland populations was limited [112], except in the Southern Cone, which is clearly visible in maps of indigenous languages as well [113]. There was, however, early dispersal of eastern lowland plants into the Andes and the Pacific coast [40,114]. In what follows, we will highlight the earliest archaeobotanical evidence of plants that were being dispersed and cultivated to show that domestication started among people generally considered hunter-gatherers [40,115].

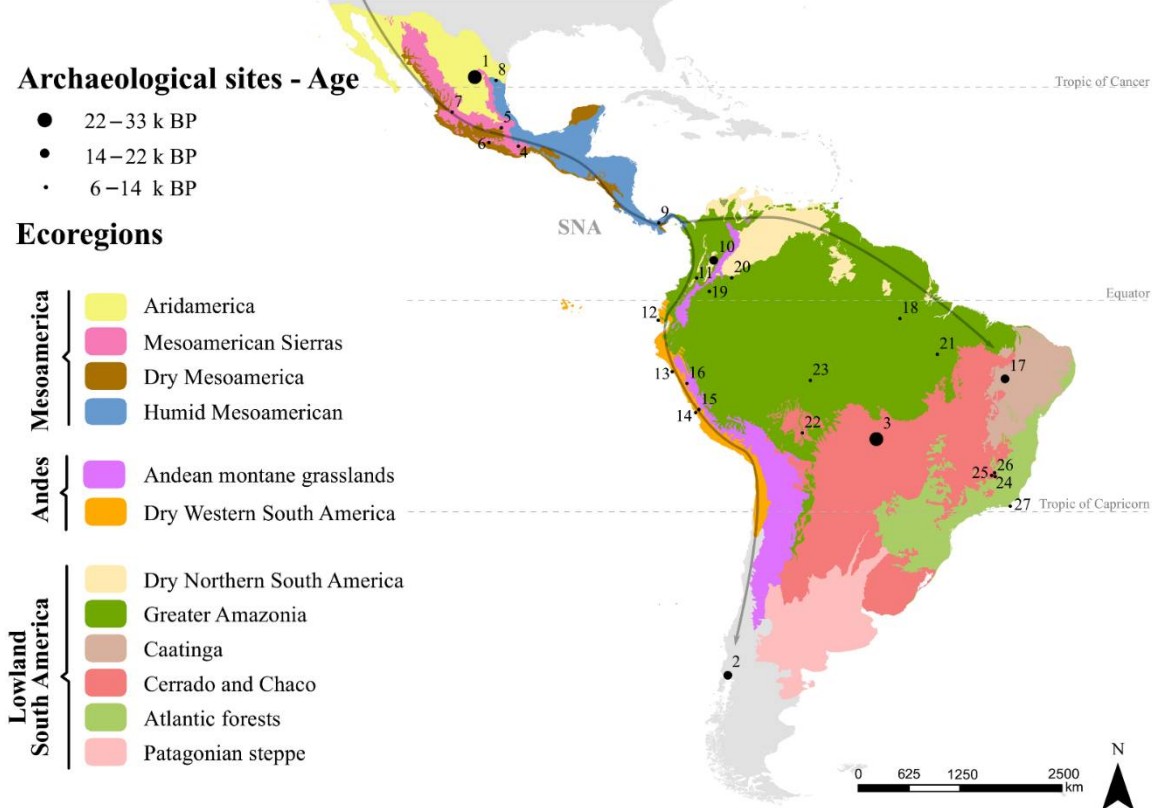

**Figure 2.** Map of the Neotropical region with major ecoregions and archaeological sites mentioned in the text. The black circles show the location of the archaeological sites; the size of the circles represents three age ranges (for correspondence of numbers and names of sites see Appendix B). Ecoregions are a combination of CONABIO [116] and Olson et al. [117] for Mesoamerica, and Romano [118] and Antonelli et al. [119] for South America. Hypothetical Southern Native American (SNA) migration routes (gray arrows) follow Waters [108].

By ~10 k BP, squash (*Cucurbita pepo* ssp. *pepo*) was being domesticated near the Guilá Naquitz cave in Oaxaca, southwestern Mexico [120]. However, domestication probably started even earlier, because the wild ssp. *fraterna* is native to northeastern Mexico [121,122]. Bottle gourd (*Lagenaria siceraria*), an exotic crop, appears by ~7 k BP in Guilá Naquitz [123]. By ~9 k BP, maize was being domesticated in the Balsas River basin of Guerrero, southwestern Mexico [124], and perhaps as early or earlier in Jalisco, western Mexico [27]. Chili pepper (*Capsicum annuum*) appears in the archaeological record by ~6 k BP in the Tehuacán valley, Puebla, south-central Mexico [125], although Pickersgill [126] considered that changes in domestication syndrome traits are only clear after ~3 k BP. This latter time frame is when two beans appear in the archaeological record (*Phaseolus acutifolius*, *P. vulgaris*) in Tehuacán [127], although *P. vulgaris* may have been domesticated in the Oaxaca River valley [128]. Squash, maize and beans are the famous triad of Mesoamerican food production systems that emerged during the Formative period, which started ~4 to ~3 k BP [123].

This quick history is biased by the standard narrative, which considers food production to be about annual crops. In Mesoamerica, perennials were as important, especially initially (see palm use during the LGM [105]). The best-known fruit crop is avocado (*Persea americana*), which appears in the archeological record by ~9 k BP in Tehuacán [128], where it was not a component of the native vegetation. Prickly-pear cactuses (*Opuntia* ssp.) appear in Tehuacán by ~13 k BP, Guilá Naquitz by ~12 k BP, and Tamaulipas, eastern Mexico, by ~10 k BP, and maguey (*Agave* ssp.) appear in Guilá Naquitz by ~12 k BP, Tamaulipas by ~10 k BP, and Tehuacán by ~8 k BP [123]. Numerous agave and cactus species are native in these areas and all are easily propagated and managed [15,61], so—like avocado—were probably being cultivated. Mesquite (*Prosopis* ssp.), with its sweet pods, appears in Tehuacán by ~13 k BP, where it is native. Slightly later, after ~7 k BP, coyol palm (*Acrocomia aculeata*), zapote blanco (*Casimiroa edulis*), zapote negro (*Diospyros digyna*) and ciruela (*Spondias purpurea*) appear in many sites [123].

Further south, in Panama, humans were using fire to manage landscapes within and adjacent to humid forests by ~11 k BP ([40] p. 209). By ~7 k BP, arrowroot (*Maranta arundinacea*) was probably being cultivated, followed shortly thereafter by leren (*Calathea allouia*) ([40] p. 213). Coyol palm, a *Scheelea* palm (possibly corozo, *Attalea butyracea*), nance (*Byrsonima* ssp.) and coubaril (*Hymenaea coubaril*) all appear by ~7 k BP [123].

The inter-Andean valleys in Colombia and the adjacent Pacific and eastern lowlands were sometimes considered centers of crop diversity by Vavilov, although this area is not included in his famous map. Piperno and Pearsall [40] accepted it and added Pacific Ecuador. In Colombia, the earliest site known is ~20 k BP at Pubenza, in the Magdalena River valley [129]. At Popayan, in the upper Cauca River valley, dated to between ~10 and ~9.5 k BP [130], starch grain evidence includes arrowroot, possibly cocoyam (*Xanthosoma* sp.), sweet potato (*Ipomoea* sp.), and manioc (*Manihot* sp.), and macro remains of tree crops, such as avocado, coyol, basul (*Erythrina* cf. *edulis*), *Caryocar* sp. and *Virola* sp., the latter introduced from Amazonia where it is used as an hallucinogenic. Gnecco [131] pointed out that the majority of these are not native near the sites, so were introduced and probably being cultivated.

Further south, Piperno and Stothert [132] identified the domestication of a local squash (*Cucurbita ecuadorensis*) starting between ~12 and ~11 k BP on the Santa Elena Peninsula, Ecuador, using changes in phytolith size through the sequence; the early domesticate was later abandoned when *C. moschata* was introduced. By ~10 k BP, leren and bottle gourd were being cultivated.

Humans certainly arrived along the Pacific coast and had explored the high elevation Andes (>3800 m) by ~13 k BP [133], although occupation was somewhat later (~10 k BP) because humans needed to adapt to hypoxia above 2500 m [134]. Along the northern Pacific coast at Huaca Prieta, in the Chicama Valley, squash, avocado and a possible medicinal plant (*Tessaria integrifolia*) appear by ~11 k BP [135], followed by chili pepper by ~8 k BP (probably *C. baccatum* [136]); only the medicinal may be native. Along the central Peruvian coast, manioc, sweet potato, potato, jicama (*Pachyrrhizus tuberosus*) and

ullucu (*Ullucus tuberosus*) appear between ~11 and ~9 k BP in the Chilca Valley and in the Tres Ventanas Cave in the mid-elevation Andes above the Chilca Valley [134]. None of these roots and tubers have wild ancestors along the coast, so they were domesticated elsewhere, introduced and cultivated. In the mid-elevation Andes north of the Chilca Valley, Guitarrero Cave had a different set of species between ~11 and ~9 k BP: common bean, lima bean (*Phaseolus lunatus*), oca (*Oxalis tuberosa*), chili pepper, and the fruit tree lucuma (*Pouteria lucuma*). The common bean and the lima bean were domesticated in both Mesoamerica and the Andes [126], but the dates at Guitarrero have been shown to be later, as they are in Mesoamerica.

At Monte Verde, adjacent to Chiloe Island, whose four cultivated species (potato, strawberry (*Fragaria chiloensis*), *Madia sativa* and *Bromus mango*) fascinated Vavilov [47], only a wild potato (*Solanum maglia*) is mentioned in the earliest archaeobotanical records, along with 45 terrestrial and numerous seaweed species as parts of the diet [109,137]. None of the terrestrial species appear to have been cultivated at that time (~14 k BP) [137].

The lowlands east of the Andes started to be occupied before ~27 k BP. The oldest occupations accepted to date include ~27 k BP at Santa Elina in the central Brazilian Cerrado [110,111], ~22 k BP in the Serra da Capivara in the northeastern Brazilian dry forest [138], ~13 k BP in the Amazon basin [139] and ~11 k BP in the southeastern Atlantic Forest [140]. These dates suggest a chronology that is the opposite of what is expected, e.g., north to south, but highlights how little we know about the occupation of lowland South America today.

Although not recognized by Vavilov, Amazonia is a center of crop genetic diversity [71], where numerous crops were domesticated and then dispersed to the Andes and Mesoamerica. At the Peña Roja site, Caquetá River, Colombia, *Xanthosoma* sp., and exotic squash, leren and bottle gourd [141,142] are considered clear evidence of plant cultivation [143] between ~11 and ~9 k BP. While annual crops appear early in Amazonia, the diversity of palms, fruit and nut trees found in those early occupations is even more significant. Phytolith and charred plant remains dating from ~12 k BP at Cerro Azul, Guaviare River, Colombia, highlight the expressive use of palms, such as assai (*Euterpe precatoria*), pataua (*Oenocarpus bataua*) and moriche (*Mauritia flexuosa*) [144]. At Peña Roja, a wide variety of palms appears between ~10 and ~9 k BP, including tucumã (*Astrocaryum aculeatum*), inajá (*Attalea maripa*), moriche, pataua, bacaba (*Oenocarpus bacaba*), and assai [141,145]. Fruit trees, such as *Brosimum* sp., aguacatillo (*Anaueria brasiliensis*), *Vantanea peruviana*, *Sacoglottis* sp. and piquiá (*Caryocar* cf. *glabrum*), also appear early [141]. At both Peña Roja and Cerro Azul, the increasing abundance of palm remains starting in the early Holocene suggest that several species were being managed [146], and some became incipient domesticates [71].

At the Pedra Pintada cave, Pará, Brazil, a great variety of palms, such as tucumã (*Astrocaryum vulgare*), moriche, bacaba, coyol, and fruit and nut trees, such as Brazil nut (*Bertholletia excelsa*), nance (*Byrsonima crispa*), coubaril, piranga (*Mouriria piranga*), *Sacoglottis guianensis*, pitomba (*Talisia esculenta*) and taruma (*Vitex* cf. *cymosa*), appear between ~12 and ~9 k BP [147–149]. In southeastern Amazonia, at Carajás, Brazil, several palms, and *Caryocar* sp., *Copaifera* sp., appear between ~10 and ~9 k BP [150,151].

Further to the south, in late Pleistocene and early Holocene occupations on two forest islands in the Llanos de Mojos savanna, Bolivia, plant remains attest to the use of manioc, squash and leren between ~10 and ~8 k BP, and maize and rice (*Oryza* sp.) by ~6.5 k BP [152]. Nearby at the Teotônio site, Rondonia, Brazil, cultigens such as manioc, squash and leren, as well as a bean (*Phaseolus* sp.), also appear in pre-ceramic occupations between ~9 and ~5 k BP [153].

In the northeastern Brazilian savanna, the analyses of pollen grains found in human coprolites dated between ~8 k and ~7 k BP, at the Boqueirão da Pedra Furada rock shelter, show the use of a great diversity of plant taxa, including *Borreria* sp., *Sida* sp., *Terminalia* sp., *Bauhinia* sp., and *Anadenanthera* sp., which could have been used as medicines. Other taxa, such as *Phaseolus* sp., Cucurbitaceae and Convolvulaceae, suggest the use of annual plants in the early Holocene diet [154,155]. Further southwest, archaeobotanical data

suggest the early cultivation of non-native species, such as sweet potato and yam (*Dioscorea* sp.), between ~12 and ~8 k BP at the Lapa do Santo cave, associated with palms [156]. Likewise, at Lapa Grande de Taquaraçu, yams and a diversity of palms appear by ~11 k BP [157,158]. Remains of pequi (*Caryocar brasiliense*), courbaril and licuri (*Syagrus coronata*) appear associated with burials of early inhabitants of the Santana do Riacho cave between ~11 and ~9 k BP [159].

In the Atlantic forest and along the Brazilian coast, archaeobotanical investigations have not yet found plant remains as old as those in Amazonia and the central Brazilian savannas. The earliest evidence comes from the Forte shellmound, on the southeastern Brazilian coast, which contains remains of yams by ~5 k BP and a variety of palms, notably *Syagrus* sp. by ~6 k BP. Other shellmounds along the southern and southeastern Brazilian coast have sweet potato, leren, Myrtaceae fruits, such as *Eugenia* sp. and *Psidium* sp., and palms, such as *Astrocaryum* sp. and *Bactris* sp., by ~ 4.5 k BP. Maize and squash appear by ~3 k BP [160,161]. Although *Araucaria angustifolia* remains have been associated with more recent human occupations (~1.5 k BP) in southern Brazilian sites [161], the expansion of Araucaria forests began by ~4 k BP, associated with human landscape management [21,162].

As in Mesoamerica, this quick survey shows the use of numerous annual and, especially, perennial species in the late Pleistocene and early Holocene, thousands of years before food production systems can be recognized in the early Formative (~4 to ~3 k BP) [40]. At many sites, species introduced from elsewhere confirm human-mediated dispersal and strongly suggest cultivation [131]. Many of the species went on to become domesticates with recognizable domestication syndrome traits. Others are currently being investigated, both in terms of their domestication syndromes and in terms of domestication processes of how human care, selection and accumulation interact with landscapes to enhance food availability.

## 4. Expansion of Plant × Human Entanglements and Domestications

In this section we quantify current plant use and management in Mesoamerica, the central Andes, and lowland South America east of the Andes, because current scenarios of plant use and management are keys to interpret the history of plant–human relationships [20]. We emphasize the numerous types of plant management that represent care, selection and accumulation, which often result in domesticated populations and landscapes. Numerous species that appear in this section also appeared in the previous section, highlighting that, although highly dynamic, the long-term interactions among humans and plants in the region show many continuities. Note that there are numerous interactions outside of food production systems per se and that a majority of these result in enhanced food availability for gathering and management.

Modern knowledge about human–plant interactions is the purview of economic botany and ethnobotany. We used three databases to quantify the state-of-knowledge about these interactions in the Neotropics and describe the human behaviors involved in landscape and plant domestication. In Mexico, the Banco de Información Etnobotánica de Plantas Mexicanas (BADEPLAM), created by Dr. Javier Caballero and currently coordinated by Dr. Andrea Martínez-Ballesté and Biol. Laura Cortés-Zárraga at the Jardín Botánico, Universidad Nacional Autonoma de México, has been compiling this information for more than 30 years [163]. In Brazil and Peru similar efforts are underway, with UseFlora since 2014, coordinated by Dr. Natalia Hanazaki, Dr. Nivaldo Peroni, Dr. Carolina Levis and Dr. Sofia Zank at the Federal University of Santa Catarina, and with the Flora Utilizada en el Perú since 2019, coordinated by Dr. Fabiola Parra-Rondinel at the Universidad Nacional Agraria La Molina. These three databases are still incomplete, so the quantification that follows is preliminary (Table 1), but is sufficient to provide a clear idea of plant–human interactions in the Neotropics. Information from the World Check List of Useful Plants [164] and the Mansfeld's World Database of Agriculture and Horticultural Crops [165] was also used to expand the list presented in Table 1 for the South America lowlands.

**Table 1.** General panorama of the native plant species used and managed as resources in the Neotropics. Notice that total numbers (#) and percentages of forms of use and management are higher than 100%, since a species may have more than one use and form of management. Because useful species are widely distributed across the Neotropics, some may overlap among regions (Mesoamerica, Central Andes and South American lowlands). Access to lists can be requested from the coordinators of each data base.

| | Mesoamerica | Central Andes | S. American Lowlands |
|---|---|---|---|
| # species used | 6500 | 1206 | 6261 |
| # families | 265 | 156 | 236 |
| Main families with (# useful species; percent **of # species used**) | Fabaceae (699; **10.8**) Asteraceae (571; **8.8**) Cactaceae (438; **6.7**) Poaceae (335; **5.2**) Euphorbiaceae (205; **3.2**) Malvaceae (171; **2.6**) Solanaceae (162; **2.5**) Rubiaceae (159; **2.5**) Asparagaceae (143; **2.2**) Apocynaceae (133; **2.1**) Lamiaceae (133; **2.1**) | Asteraceae (107; **9.4**) Fabaceae (88; **7.3**) Solanaceae (84; **7.0**) Rubiaceae (32; **2.6**) Poaceae (28; **2.3**) Rosaceae (28; **2.3**) Euphorbiaceae (22; **1.8**) Cactaceae (21; **1.7**) Lamiaceae (21; **1.7**) Amaryllidaceae (19; **1.6**) Apiaceae (18; **1.5**) | Fabaceae (840; **13.4**) Euphorbiaceae (262; **4.2**) Rubiaceae (227; **3.6**) Asteraceae (223; **3.6**) Malvaceae (183; **2.9**) Lauraceae (182; **2.9**) Poaceae (179; **2.8**) Myrtaceae (172; **2.7**) Apocynaceae (162; **2.6**) Annonaceae (154; **2.4**) Solanaceae (151; **2.4**) Arecaceae (138; **2.2**) |
| Main uses with (# species; **percent** of # species used) | Medicinal (3478; **53.5**) Edible (1810; **27.9**) Fodder (1637; **25.2**) Construction (1224; **18.8**) Fuel (883; **13.6**) | Medicinal (644; **53.0**) Edible (424; **34.9**) Environmental (315; **30.9**) Manufacture (229; **18.8**) Construction (169; **13.9**) | Medicinal (4017; **64.1**) Manufacture (2175; **34.7**) Edible (1719; **27.4**) Construction (1683; **26.8**) Fodder (667; **10.7**) |
| Habit with (# species; **percent** of # species used) | Herbs (2619; **40.3**) Trees (1861; **28.6**) Shrubs (1411; **21.7**) Lianas (499; **7.7**) | Herbs (567; **47.0**) Shrubs (339; **28.1**) Trees (248; **20.5**) Liana (52; **4.3**) | Trees (3319; **53**) Herbs (1276; **20.4**) Shrubs (1002; **16**) Lianas (651; **10.4**) |
| Species gathered | 6000 (**92.3%**) | 778 (**64.5%**) | 6178 (**98.6%**) |
| Species managed | 1555 (**23.9%**) | 428 (**35.5%**) | 742 (**11.8%**) |
| Domestication | | | |
| Incipient | 727 (**11.2%**) | Not available yet | 517 (**8.2%**) |
| Semi | 170 (**2.7%**) | 304 (**25.2%**) | 38 (**0.6%**) |
| Full | 251 (**3.9%**) | 124 (**10.3%**) | 45 (**0.7%**) |

### 4.1. Mesoamerica (Mexico and Central America)

In Mexico and Central America botanists have recorded 39,304 species of vascular plants [166]. In Mexico alone, the inventory is 23,314 species [167]. Ethnobotanists have recorded nearly 6500 native species of useful plants (about 20% of all species recorded for the area) (Table 1), most of which (6000 (~92%)) are wild, obtained from forests and other ecosystems by simple gathering to satisfy a variety of needs, mainly medicine (~3480 species) and food (~1812 species) [168]. Among the 6000 species obtained through gathering, 1555 (24%) are managed with different silvicultural and horticultural practices in their native ecosystems. For well-studied areas, such as the Tehuacan Valley, ethnobotanical studies have documented that ~38% of ~2000 species used by people are under some form of management [61].

The most common form of management for the 1555 managed species is to tolerate and protect a plant when areas are cleared for food production plots, pastures and other purposes, which occurs with 855 species. This form of management includes weedy plants that are selectively tolerated and protected in food production plots during weeding, mainly because they are valuable edible plants, especially greens (generically called "quelites" in

Mesoamerica) and several species of "tomate" of the genera *Solanum*, *Jaltomata* and *Physalis*. In most cases, this diversity becomes part of agroforestry systems. People report recognizing and naming phenotypic variants among these protected species [15,60,61,169]. Such recognition is relevant because the named phenotypes have different attributes that are also differently valued by people. For instance, people recognize "quelites" with good texture and flavor (commonly called "female" plants) and distinguish them from others with more rigid texture and bitter flavor (commonly called "male" plants); furthermore, people distinguish varieties of small and large "tomates" [60]. This distinction leads to the selection favoring desirable phenotypes, increasing their frequencies in agroforestry systems or in different aged fallows, and thus represents traits of the domestication syndrome [15].

Among the tolerated/protected species with recognizable phenotypes, 163 species are also managed to enhance their abundance. The most common practices recorded are planting their seeds and/or vegetative propagules, or even irrigating or firing areas where they occur. The latter practice is common for those species that are tolerant of fire and this activity favors their abundance; an example is the palm *Brahea dulcis* [14,61,90,107,170,171]. With an additional 170 species, people practice special care, such as removing competitors or herbivores; some plants are shaded when clearing forest or, for other plants, branches of neighboring plants are removed to reduce shade that limits their growth, as is the case of some *Agave* species [172].

Seven 127 species are recorded as being transplanted from forests to anthropogenic areas (incipient domestication; Table 1). The direct propagation of plant propagules is the most common form of plant management, but the transplanting of complete individual plants occurs with 10% of managed plant species (~130 species) and sowing of seeds occurs with 25% of managed plant species (~320 species). In ~251 plant species, a combination of human selection, sowing of seeds, vegetative propagation, and transplanting of entire plants has been recorded. These practices are clearly part of the domestication process and have been documented for numerous species of cacti, *Agave*, trees, shrubs and epiphytes [15,61–63,170–177].

The cultural importance of a plant resource is the main factor influencing management intensity. This includes the reasons why it is appreciated by people and its role in subsistence, either for direct consumption or for its exchange value. Ecological factors that influence the resource's availability may influence management intensity. A culturally valuable resource that is naturally scarce, with a small distribution, or whose availability is uncertain because it is vulnerable to inter-annual climate changes, often causes people to invest more effort to ensure its availability. Some biological features of a plant may influence how feasible it is to manage. Among the most common difficulties are slow growth and length of life cycle, seed dormancy, specialized ecological interactions or habitat conditions for establishment. In contrast, ease of germination and establishment, vegetative propagation, fast growth and reasonable time to harvest may favor management intensity. As such, the typology of management categories, with their respective practices, represents a continuum of management intensity [61,62,170,171]. Although these observations are from Mexico, the human behaviors and practices involved are common across the Neotropics (and the world) and can be hypothesized to have occurred at least during the Holocene, if not earlier.

Management intensity is a combination of the practices involved, human selection and its intensity, the quality and quantity of products obtained, and the destination of these products. In Mesoamerica, species grown strictly from seed and that have slow growth receive limited management, even when their fruits are appreciated. In the cases of columnar cacti, these species are mainly gathered from wild populations and protected where the forest is cleared. People use and protect plants producing larger and sweeter fruits, as well as those with special traits, such as fewer or shorter spines, and pulp colors other than red, which is the most common in the wild. In contrast, species that are vegetatively propagated are also dispersed and propagules are cared for [15]. Among these species, *Polaskia chichipe* and *Myrtillocactus schenckii* are dispersed in farming plots

and homegardens. Another group of species, represented by *Stenocereus stellatus* and *S. pruinosus*, are easily vegetatively propagated and grow much faster than in *P. chichipe* and *M. schenckii*. These species are much more common in homegardens, live fences, and there are plantations for producing fruits [18,178]. Human selection on these populations has resulted in changes in morphology, genetic diversity and structure, reproductive biology and germination patterns—all components of their domestication syndromes. With other species, such as *Leucaena esculenta*, *Crescentia cujete* and *C. alata*, and *Sideroxylon palmeri*, people select for larger fruit size [15,179–181]. In the case of *Agave*, people favor larger plant size, since stem biomass and leaf length are important to produce sap for fermentation and fibers, respectively [182]. Additionally, in agaves, saponins that irritate human skin have been reduced or eliminated during domestication to make management easier [183]. In *Bursera bipinnata*, management has increased the abundance of phenotypes producing more resin with aromatic compounds appreciated by people [184]. In all cases, morphological and genetic divergence between wild and managed populations exists and is proportional to management intensity.

### 4.2. Central Andes

The Central Andes, a region that has the highest and longest segment of the Andes [185], has a vascular plant richness of 17,548, 19,147, and 14,431 species in Ecuador, Peru and Bolivia, respectively [166], with considerable overlap. Our focus will be restricted to the Andes above 500 m on the western slopes [186] and above the "eyebrow" of the Selva on the eastern slopes [187]. The distribution of the species we include in Table 1 was verified with the Catalogue of Plants of Peru [188], as well as with the Plant List.

Several attempts have been, and are being, carried out to systematize lists of useful species in these countries. In Peru, the most important compilation was carried out by Brack-Egg [189] in his Encyclopedic Dictionary of Useful Plants of Peru, which is the starting point of our first approach to systematize information on the useful plants of the central Andes. Brack-Egg [189,190] identified approximately 4400 useful native species in all Peru, almost a fifth of the 19,147 Peruvian plant species [166]. We add about 90 species from Pancorbo-Olivera et al. [191], which provides a detailed record of how Andean communities in central Peru handle plants for food. From this universe, we identified 1206 species as Andean, most of which have medicinal uses (644 species), food uses (424 species), and environmental functions (315 species). Of these species, 778 are gathered from the wild. Another 304 species were categorized as semi-domesticated and 124 species as domesticated; both categories include species that are cultivated with various types of sexual or asexual propagules, and are also collected from wild populations of the domesticated species.

Although the dominant life form characteristic of high Andean ecosystems is herbaceous (567 species; Table 1), the number of perennials (mainly trees and shrubs, 587 species in total) is considerable. The perennials are characteristic of the Andean forests, such as the montane or yungas of the tropical Andes [185], which become premontane below 1000 m and gradually grade into the "eyebrow" of the Selva [192]. These forests contain great diversity, although this is little studied and vulnerable to degradation, and, together with other forests, such as the Peruvian relicts [193,194], equatorial and dry forests of the inter-Andean valleys, contribute to the list of perennial useful species. The study and conservation of Andean forests are key issues, because they represent a source of useful species that complement human needs not fully covered by domesticated species [191], as well as being the habitats of many wild relatives of useful species [195]. These gene pools have contributed and continue to contribute to the diversity of domesticated populations at the intraspecific level.

The most representative and well-known domesticated plants of the Central Andes are the tuberous species, such as potato (*Solanum* spp., locally known as *papa*), oca, ulluco, and mashua (*Tropaeolum tuberosum*), which are important for their nutritional contribution to the region and the world. In many Andean food production systems it is still possible

to find these crops coexisting with their wild relatives, called *atoq* (fox) or *k'ita* (from the wild or the mountains) in Quechua. *K'ita* are mainly found on the edges of cultivated fields (called *pirqas*) and in neighboring plant formations, especially shrublands and forests [195]. In some cases the remnants of previous harvests (called *k'ipas*) are frequent [196]; "*k'ipas* is a Quechua term referring to volunteer plants of a crop species emerging in sites of former fields in the years following cultivation of this crop" [197]. Sometimes Andean farmers identify new varieties derived from *k'ipa papa*, which suggests that these may result from seeds and, therefore, sexual reproduction that may involve crosses among wild, weedy and cultivated variants [196]. Similar situations have been documented for mashua [198] and oca [197].

In the southern Peruvian Andes it is very common to find *Araq papa*, considered to be *S. tuberosum* subsp. *andigena*, and a group of poorly studied and unclassified taxa that behave like weedy plants and seem to have considerable intraspecific diversity; their root systems typically have a long stolon and thick tuber skin [196]. This sub-species and other taxa are collected by local farmers and can be an important source of food [199], especially during the period before the harvest of cultivated potatoes, and they are sold in local fairs. One estimate suggests that nearly one third of the total annual consumption of Andean tubers by regional people is provided by *araq papas* [200].

Sometimes these *araq papas* are grown near plots of other potatoes by clonal propagation with the intention to conserve variety or acquire new tasty varieties. This coexistence contributes to natural gene flow and the appearance of new varieties of potato. Reproductive interaction and gene flow by natural pollinators among wild and cultivated potatoes has frequently been documented [201–205] and studied locally by Marquez-Castellanos et al. [206]. From this gene flow, new varieties arise as *k'ipas*, sexual hybrids that can be included in the complex array of landrace cultivars; thus, they represent an important source of phenotypic variability available for selection by farmers. More studies are needed about the current processes of the generation of crop diversity to understand how this occurred in the past and in the present to create the high levels of intraspecific variability of modern tuberous crops, and in this way help to understand the entangled origins of many crops in the Andean region.

*4.3. South American Lowlands*

In South America, botanists have recorded 82,052 species of vascular plants [166]. Brazil has the most diverse flora within South America, with 33,161 species [166]. Ethnobotanists have recorded at least 6261 native useful plants gathered from lowland forests and other ecosystems to date, with the Fabaceae, Euphorbiaceae, Rubiaceae and Asteraceae families the most frequent (Table 1). A majority of these 6261 species are used for medicine (4017 species), cultural activities and manufacture (2175 species), and food (1719 species). Among these useful species, at least 742 species (11.8%) are managed in their ecosystems or in domesticated landscapes, 594 species (9.5%) are dispersed and propagated by humans, and 585 species (9.3%) are promoted by soil improvement and fire management activities. At least 600 (9.5%) of all the useful plants are cultivated or have some evidence of selection and propagation (therefore domestication). There is considerable overlap of useful species among the biomes, with 4216 out of 6261 species known to be used in Amazonia, 2913 species in the Atlantic Forest, 2384 in the Brazilian Cerrado and other savannas, 1463 in the Brazilian Caatinga and possibly other seasonally dry forests, and 662 in the Pampas and 633 in the Pantanal. These uses and management are a result of different cultures, knowledge, practices and ecosystems, and have led to different expressions of landscape and plant domestication across these biomes.

In Amazonia, 2253 useful tree and palm species (nearly 50% of all known arboreal species in the region) belong to 100 botanical families [207]. Together these useful species represent 84% of all the trees and palms estimated to occur in Amazonian forests, because a majority of the 227 hiperdominant species are useful [207,208]. On average these useful trees and palms have population sizes six times larger than non-useful species [207].

Among the useful tree and palm species, 45 native species have incipiently domesticated populations, such as Brazil nut, assai (*Euterpe precatoria* and *E. oleracea*) and rubber (*Hevea brasiliensis*), and are the most abundant species in these forests [207]. Amazonian societies may have favored the abundances of these species in present-day forests [102]. Amazonian peoples practice diverse cultural management activities to care for plants (generically called "*zelos*" in Brazil), which are responsible for the formation and maintenance of Amazonian domesticated forests [67]. These practices were used more intensely in the vicinity of archaeological sites (up to 4 km) and sometimes current settlements [209,210], where aggregations of useful and domesticated species are common. In the lower Tapajos River region, for example, there are large piquiá (*Caryocar villosum*) stands near archaeological sites and local communities have selected and propagated their preferred types into home-gardens [211]. At least four genetic groups have been identified, showing an accumulation of diversity, while also presenting lower than expected genetic diversity, which suggests incipient domestication of this forest emergent [212]. For fully domesticated populations, morphological and genetic differences are more pronounced, as demonstrated by peach palm (*Bactris gasipaes*), which often presents seedless fruits, and some landraces have fruits up to two hundred times heavier than the wild populations [213].

In the Brazilian Cerrado, fire has been used as a management tool for millennia, as is typical in heterogeneous savanna landscapes in the Americas and elsewhere [214]. Setting fields on fire is a practice that favors the flowering and fruiting of numerous species of interest, such as capim-dourado (*Syngonanthus nitens*), used for making handicrafts [215]. In addition, some species are protected from fire to continue providing human resources, such as buriti (*Mauritia flexuosa*) [215]. Fire also promotes the re-growth of grasses that attract game animals and prevents severe forest fires by eliminating excess combustible material [216,217]. In Central Brazil, pre-Columbian management of plants, such as pequi (*Caryocar brasiliense* and *C. coriaceum*) and janaguba (*Himatanthus drasticus*), resulted in incipiently domesticated populations [218–220]. The Kayapó people enrich the forests adjacent to their fields, with some forest patches having up to 75% of the useful species intentionally planted [221]. Some of the species planted by the Kayapó today, such as *Spondias* sp. and *Hymenaea* sp., are found in archaeological sites in other parts of the Brazilian savanna [222]. In the upper Xingu River, *C. brasiliense* has great cultural and food importance for the Kuikuro indigenous peoples, who select and manage the species in their agroecosystems [219]. In the Chapada do Araripe, northeastern Brazil, local farmers protect trees of *C. coriaceum*, and also cultivate the species after breaking seed dormancy [220,223].

In the semi-arid Caatinga, management favors perennial species with edible fruits, such as umbu (*Spondias tuberosa*) [224]. Medicinal and other food resources include trees, such as aroeira (*Myracrodruon urundeuva*), imburana (*Amburana cearensis*), juazeiro (*Sarcomphalus joazeiro*), and species of Cactaceae (*Cereus jamacaru*, *Melocactus zehntneri*, and *Pilosocereus pachycladus*), which present evidence of tolerance, protection and vegetative propagation [225,226]. The Caatinga is also home to *Neoglaziovia variegata* (Bromeliaceae), which has been cultivated and domesticated for its leaf fiber [71]. The cashew (*Anacardium occidentale*), widely appreciated and cultivated for its edible pseudo-fruits and nuts, was probably domesticated in seasonally dry forests of northeastern Brazil, where the greatest diversity of cultivated varieties is found [227]. As elsewhere, this human management influenced both species and landscapes to meet different human needs [228]. The Caatinga is the least studied of the Brazilian biomes [229], so there are certainly many more species under domestication to be studied in this region.

In the Atlantic Forest of southern and southeastern Brazil, the Tupi-Guarani and Macro-Jê linguistic families were particularly important along the coast and in the interior, respectively, where they used and managed at least 29 tree species commonly found in present-day forest fragments [230]. Starting ~4 k BP, araucaria (*Araucaria angustifolia*) expanded through the gallery forests in the interior of the southern Atlantic Forest [162]. Between ~1.4 and 1 k BP, this was mainly driven by human dispersal [231]. Before the arrival of Europeans in South America, the Araucaria Forest occupied an estimated area of

200,000 km$^2$ in Brazil and Argentina [232], and yerba mate (*Ilex paraguariensis*), pineapple guava (*Acca sellowiana*), butiá (*Butia eriospatha*), caraguatá (*Bromelia antiacantha*), and several Myrtaceae species were being managed in this landscape as well [21,233]. Yerba mate is used and intensively managed to make a traditional tea-like beverage in Brazil, Argentina, Uruguay, and Chile, and is used as a medicinal plant. In the case of araucaria, ethnobotanical studies with local farmers described several ethnovarieties [234], and some of these are classified as botanical varieties as well [235,236], supporting its classification as at least an incipient domesticate. In the last two centuries, these forests have been managed by traditional people who associate subsistence cultivation and animal husbandry with the extraction of native species, such as araucaria and yerba mate, in systems locally known as "*faxinais*" or "*caívas*" [21,237]. These systems form a mosaic of communal areas composed of more intensive cultivation areas, pastures, and forest fragments, which contribute to the conservation of araucaria forests and local cultural traditions. The Araucaria Forest can be understood as a mosaic of highly productive cultural landscapes, which facilitate the co-occurrence of other useful species at the landscape level that have also been managed and domesticated, as well as being responsible for the maintenance of the remaining native vertebrates [72,73,238].

*4.4. Comparisons among Regions*

The three Neotropical regions we focus on show important differences in useful plant species richness (Table 1). This is partly due to natural differences in plant diversity among the ecosystems occurring in these regions. However, it is necessary to emphasize that the information reflects different efforts invested in the research and systematization of information in these regions. BADEPLAM has dedicated more than 30 years to systematizing information in Mexico, which is one of the countries with higher plant and cultural diversity, and where ethnobotanical research has been active [239]. The Useflora database has systematized information from a wide area with the highest biocultural diversity of the Americas, but has been working for only five years. The Plantas Utilizadas de Perú database of Andean ethnobotanical information is a very recent effort in an area with ancient cultures and high biocultural diversity, with enormous demands for exploring and studying numerous important areas.

In general, 60% of the plant families providing higher numbers of useful plants are similar among regions. Fabaceae and Asteraceae are the most frequent families in all regions. There are also clear differences, with agaves and cacti more important in Mexico than in any other region, and with the Rosaceae, Apiaceae and Amarilidaceae standing out in the Andean region, while Lauraceae, Myrtaceae, Annonaceae and Arecaceae are more important in the lowlands than in any other region analyzed. It is not only the diversity and abundance that influence the salience of these families, but the quality of their products to satisfy human needs.

The numbers of useful plant species reported in the three regions reveal that resources from forests continue to be important for human subsistence and health, and different forms of management increase their availability and/or quality. This was certainly true in the past too, perhaps even more so. Even so, herbaceous plants are particularly important in Mesoamerica and the Andean regions, whereas trees are more important in the lowlands, which is certainly due to the highly diverse forests of Amazonia and the Atlantic Forest.

The domestication process generates larger responses and in less time in annuals and other short life-cycle plants than in long-lived perennials. This is probably the reason why full domesticates are more common in Mesoamerica and the Andes than in the lowlands. In addition, research in Amazonia has suggested that arboreal food species can provide more food than expected, so may reduce the importance of intensively cultivated food production systems [37,103,240]. This may explain the number of species with incipiently domesticated populations in contrast to the species with semi- and domesticated populations [71].

## 5. Are the Neotropics Different?

The Neotropics did not have a Neolithic Revolution as described for southwest Asia [37,40,41,103], which places the region in good company, as the various regions of the Old World where states arose and collapsed did not either [3]. As Piperno and Pearsall [40] point out, all regions that saw the rise of states are different to some extent, especially in the details of what was domesticated and when, and how these plants and animals were integrated into the food production systems upon which each state relied. We can continue disentangling the management of plants and animals and, eventually, their domestication *sensu* Darwin [23] from reliance on food production systems by looking at the diversity of relationships among plants and Neotropical societies.

The Neotropics are extremely diverse ethnically, which is measurable as language diversity. According to Ethnologue [241], Mexico and Central America currently have 327 languages, while South America has 455 languages. The current language richness is a small proportion of that existing before the depopulation caused by European conquest [242].

Only a small number of these ethnic groups expanded more than others, and an even smaller number gave rise to the hierarchical societies called states. At the time of European conquest, there were no reports of any groups living with hunger, so we can assume that all of these large and small ethnic groups had access to sufficient plant and animal resources in their territories. The earliest European reports did not mention groups without some kind of food production, although these existed in arid and semi-arid northern Mexico [243], and may have existed far from major rivers and coasts in lowland South America where semi-nomadic groups exist today, such as the Nukak, who had no food production systems but created forest orchards by other means [244].

Since plant and landscape domestication are the consequences of very basic human behaviors and associated practices, we suggest that the majority of these ethnic groups managed the plants that most interested them, sometimes only as a result of landscape domestication, which can result in incipient changes in the domestication syndrome [15], and which Rindos [5] called incidental domestication. As we have shown, the number of plant species with incipiently domesticated populations in the Neotropics is much larger than previous reports of plant domestication in the region. Their domestication syndromes are incipient, but visible when studied carefully [18,178,245].

All of these ethnic groups interacted with their neighbors. The exchange of preferred plants and associated techniques is a common part of these interactions [17,246], so useful plants were dispersed, sometimes attracting more attention from a neighboring group than in the group that first became interested in the species, e.g., peach palm [213]. As we have shown here, these dispersals have been recorded since the end of the Pleistocene and probably happened earlier as well, although they are more elusive in the archaeological record. Ethnographic and ethnobotanical studies have documented how these processes are currently happening, and we feel that ethnographic projection in this case is possible because of the basic human behaviors involved. These very early records of dispersal and probable cultivation are expected within the cultural niche construction framework [17,115], leading directly to low-level food production [46], and suggest that the idea of centers of origin of agriculture do not exist, as also suggested by Langlie et al. [247].

Some ethnic groups accumulated more species than others, which required caring for them and often required cultivation. This initiated the entanglement of specialized domestication and the creation of agroecologies, initially of low-level food production systems [46]. Among some ethnic groups, these low-level systems developed into more complex and more intensive systems, without abandoning their use of other ecosystems, especially forests. Even today in Mexico, forest foods contribute 10-20% of the annual diet [90,248,249]. Among other ethnic groups, the low-level systems did not become more intensive. The Huaorani of Ecuador had only three cultivated species, until forced to settle by modern government policies [250]: sweet manioc, peach palm and banana (*Musa* ssp.). Both native species were cultivated exclusively for annual festivities. All of the rest

of Huaorani food and other necessities were procured from their domesticated forests. In many parts of lowland South America this pattern of low-level food production was common, but is gradually disappearing as these small ethnic groups are forced to settle.

Throughout the Neotropics, some ethnic groups expanded their food production systems, which resulted in population expansion and the rise of hierarchy. In both Mesoamerica and the Andes, social hierarchy on the scale of states arose and collapsed, as happened in the Old World. As the scale of food production expanded, so did the accumulation of new varieties of preferred plant species and new species from other parts of the Neotropics. As we saw, this started very early and finally resulted in the centers of crop genetic diversity described by Vavilov in Mesoamerica and the Andes [47]. It is important to recognize that these are centers of accumulation [51], not only of origin, and especially not of the origin of agriculture, as they arose much later than the origins of domesticated plants. As recognized by Vavilov [48], parts of lowland South America also contained an abundance of crop genetic diversity. A mosaic of centers, regions and micro-centers was proposed [50], each in areas where human population densities were high and social hierarchy was evident [251]. The empty spaces in the mosaic are not evidence of an absence of crop genetic resources, but an absence of research to identify them among these small-scale ethnic groups. Thus, this mosaic reflects the different socio-ecological histories of the numerous small and large language families and language isolates [103,252].

In areas that naturally contained forests, early European observations mention networks of villages surrounded by mosaics of food production systems interspersed with forest [94,253], some of which represent low-density urbanism [254]. The Maya forests are the most famous and contain evidence of enrichment with useful species [98]. It is now clear that these forests were extremely dynamic, with areas cleared for milpas, which were then managed to become agroforestry systems that continued to accumulate useful species until they were mature secondary forests that supplied construction and manufacturing raw materials, as well as food and medicinals [98]. Similar systems were practiced in the semi-arid Tehuacan Valley, one of the areas where early food production systems were identified by MacNeish [255]; these systems are still practiced in different vegetation types of the region [14,15,256–260]. These highly productive mosaics of low-density urbanism also explain why there are so many useful tree and shrub species, including cacti and agaves, in our lists in Mesoamerica and lowland South America.

In Amazonia, similar palaeoecological and archaeological evidence around numerous sites suggests that this pattern was common [93,254,261]. The majority of the sites studied by these authors contain Amazonian dark earths (ADE), which are anthropic soils that originated in dump heaps [262]. The abundance of ADE sites across Amazonia suggests that this pattern is widespread in the region, including in interfluvial areas far from the major rivers (see maps in [263]). The expansion of araucaria forests in the southern Atlantic forest appears to be a less intensive manifestation of the same pattern [21,162,264].

The Central Andes were somewhat different, because they are an extremely complex mosaic of different environments [265–267], where the higher areas often have steep slopes, little or abundant rainfall, high erosion potential, etc. Anthropologists and biologists distinguish several zones: the "*selva*" (tropical wet forest), with high and low *selva* [268], the latter grading into the lowlands; the "*yungas*", which is a large biogeographic region with southern and northern differences [269] that grade into high *selva*, with tropical dry forest in some areas and cloud forest in others; the "*quechua*", with temperate latifoliate forests that grade into the *puna*; the "*puna*", with small trees and shrubs and extensive highland grasslands, where the Andean tubers prosper (potatoes, mashua, oca etc.); and the western slopes descend to the dry coast, where there are dry forests in the north with cacti and xeric vegetation [186]. This dry fringe is cut at intervals by rivers that form oases where tropical plants flourish, and which supported several early states, such as Caral and Chavin, which occupied territories with ample zonal representation. The occupants of all these zones interacted continually. The eastern *yungas* and coastal oases had food production systems similar to many in Mesoamerica, while the *selva* grades into Amazonia

with its *chagra* agroforestry systems. Where rainfall was limited, irrigation was used; where slopes were steep and erosion active, terracing of many types was used [12], as in parts of Mesoamerica. In the *puna* and upper *quechua*, annuals were dominant, while trees increased in abundance in the lower *quechua*, *yungas* and *selva*. As in Mesoamerica and the lowlands, high crop diversity was the primary means of pest control, as well as enriching diets. Within such a complex ecological mosaic, species from different zones were continually experimented with in others, with specific varieties adapted to wetter–drier or high–lower niches.

In both Mesoamerica and the Andes, the rise of states reduced the availability of land for ethnic groups that did not want to participate in the states. In Central America and, especially, in lowland South America the lack of states left space for social diversity, even for ethnic groups that did not rely on food production systems, although they did domesticate landscapes, even forests [67]. This great social diversity is the primary evidence that the domestication of plants does not lead necessarily to reliance on food production systems. Each ethnic group decided how many plants to manage or cultivate, how much reliance on food production systems was appropriate, and how to domesticate their forests and other landscapes for the rest of their sustenance. Looking in the spaces between states highlights that the standard narrative was written by the states, not by the thousands of other ethnic groups with alternative ontologies and management strategies.

## 6. Post-Script

Scientists like to finish an analysis such as this by relating it to our current predicament—the Anthropocene. What we have shown is that it is perfectly possible to live well without agriculture, in the original sense of the term, and even better without industrial agricultural systems with their high impact on global ecosystems. Imagine rolling back the industrial monocultures that feed global industrial society and replacing them with low-density urbanism based on a mosaic of food production systems, agroforests and domesticated landscapes. This image requires human labor, not machines. Large numbers of Neotropical slum-dwellers are recent migrants from rural areas where they lived in such mosaics. Valuing their knowledge, their traditions and giving them rights to their ancestral lands could roll back the Anthropocene and still feed humanity.

**Author Contributions:** Conceptualization, C.R.C. and A.C.; validation, A.M.-B., L.C.-Z., F.A.P.-R., N.H., N.P. and C.L.; data curation, C.L., N.P., N.H., G.L., L.C.-Z., S.R.-L., J.B., R.P.A., S.D.C., A.C.-S., M.P.-O., M.J.F., E.L.-V., V.M.B. and G.G.M.; writing—original draft preparation, C.R.C., A.C., M.F.C., C.L., N.P., R.P.A., N.H., S.D.C. and M.J.F.; writing—review and editing, C.R.C., A.C., S.R.-L., C.L., N.P., M.J.F., R.P.A., M.F.C., S.D.C., N.H. and G.L.; visualization, M.J.F., M.F.C. and S.R.-L. All authors have read and agreed to the published version of the manuscript.

**Funding:** This research received no external funding.

**Institutional Review Board Statement:** Not applicable.

**Informed Consent Statement:** Not applicable.

**Data Availability Statement:** Full data may be requested from the coordinators of each of the data banks (BADEPLAM, Useflor, Flora Utilizada en el Perú). E-mail addresses are in affiliations above.

**Acknowledgments:** We recognize the extraordinary effort of Javier Caballero to design, create and develop BADEPLAM, and we dedicate this study to his memory. The Conselho Nacional de Desenvolvimento Científico e Tecnológico (CNPq) provided research fellowships to C.R.C. (PQ 303477/2018-0), N.H. (PQ304525/2019-1), N.P. (PQ306801/2019-1), a doctoral scholarship to M.F.C. (169800/2018-0) and an undergraduate scholarship to G.L. (PIBIC-2019-2020); the Consejo Nacional de Ciencia y Tecnología, Mexico (CONACYT) provided a postdoctoral fellowship to S.R.-L. (A1-S-14306) and a Master's scholarship to E.L.V. (925661); the Fundação de Amparo à Pesquisa do Estado do Amazonas (Fapeam) provided a doctoral scholarship to R.P.A. (062.01758/2018) and a project fellowship to V.M.B. (062.00148/2020); the Coordenação de Aperfeiçoamento de Pessoal de Nível Superior (CAPES) provided a doctoral scholarship to M.J.F. (88882.436667/2019-01) and a postdoctoral fellowship to C.L. (88887.474568/2020); the Fundação de Amparo à Pesquisa do Estado

de São Paulo (Fapesp) provided a postdoctoral fellowship to G.G.M. (2019/07773-1). The CNPq (435985/2018-3) and Fapeam (062.00148/2020) supported field work in Amazonia; the Dirección General de Asuntos del Personal Académico (DGAPA, UNAM, proyecto IN206520) provided general support for the Mexican team. BADEPLAM thanks Francisco Mora and Hada C. Montes, Useflora thanks Patricia A. Ferreira, Beatriz Rosim, Mario Tagliari, Ariane Saldanha de Oliveira, Michel Omena and Bianca Morais, and Flora Utilizada en el Perú thanks Hernan Cunza for additional assistance with the databases.

**Conflicts of Interest:** The authors declare no conflict of interest.

## Appendix A

In general, each is tailored to the objectives of the author's publication, since there is no consensus definition. Melinda Zeder's [56] review of definitions is especially interesting.

**Table A1.** A non-exhaustive list of definitions of domestication by archaeologists, geneticists and other students of domestication since the turn of the millennium.

| Definition | Author (Year) |
|---|---|
| Domestication is a co-evolutionary process that occurs when wild plants are brought into cultivation by humans. | Purugganan [270] |
| Domestication is the process of heritable genetic adaptation to human cultivation and consumption conditions. | Hufford et al. [271] |
| Domestication is an evolutionary interaction where a producer species gains new dispersal mechanisms while its performance is controlled for the benefit (commonly nutritional) of a consumer species. | Milla et al. [272] |
| Domestication can be generally considered a selection process for adaptation to human agroecological niches and, at some point in the process, human preferences. | Larson et al. [273] |
| Domestication is an evolutionary process driven by natural and human (whether conscious or unconscious) selection applied to wild plants or animals and leading to adaptation to cultivation and consumption or utilization. | Gepts [274] |
| Domestication describes genetic and morphological changes on the part of a plant population in response to selective pressures imposed by cultivation. | Fuller and Hildebrand [275] |
| These sustaining crop plants were derived, in most cases, by several thousand years or more of conscious as well as unintentional human selection, in the process transforming mostly unremarkable wild ancestors into high-yielding and otherwise useful domesticated descendants. | Olsen and Wendel [276] |
| Here, "domesticated" refers more generally to plants that are morphologically and genetically distinct from their wild ancestors as a result of artificial selection, or are no longer known to occur outside of cultivation. We define "semidomesticated" as a crop that is under cultivation and subjected to conscious artificial selection pressures. | Meyer et al. [25] |
| "Domesticated species" are those that have been genetically altered through artificial selection such that phenotypic characteristics distinguish them from wild progenitors. | Piperno [277] |
| Domestication is the outcome of a selection process that leads to plants adapted to cultivation and utilization by humans. | Brown et al. [278] |
| Domestication is generally considered to be the end-point of a continuum that starts with the exploitation of wild plants, continues through the cultivation of plants selected from the wild but not yet genetically different from wild plants, and terminates in the fixation, through human selection, of morphological and hence genetic differences distinguishing a domesticate from its wild progenitor. | Pickersgill [279] |
| Domestication is most often defined in terms of two salient characteristics: first, that the newly created "species" is observably distinct from its wild relatives; and second, that without continued human protection, it would cease to exist. | Smith [280] |
| Domestication is best viewed as an evolving of mutualism between humans and populations of plants or animals. | Zeder [56] |
| In scientific usage, "domestication" has come to mean the process by which humans transformed wild animals and plants into more useful products through control of their breeding. | Leach [58] |
| By a domesticate, I mean a species bred in captivity and thereby modified from its wild ancestors in ways making it more useful to humans who control its reproduction and (in the case of animals) its food supply. | Diamond [281] |

## Appendix B

**Table A2.** Correspondence of archaeological site numbers in Figure 2 with site names and references cited in the text.

| No. | Name | References Cited |
| --- | --- | --- |
| 1 | Chiquihuite Cave | Ardelean et al. [105]; Becerra-Valdivia and Higham [282] |
| 2 | Monte Verde | Dillehay et al. [109,137] |
| 3 | Santa Elina | Vialou et al. [110]; Scheel-Ybert and Bachelet [111] |
| 4 | GuiláNaquitz Cave | McClung de Tapia [123]; Smith [130] |
| 5 | Tehuacán Valley | Smith [128]; McClung de Tapia [123]; Debouck [126] |
| 6 | Guerrero | Piperno et al. [124] |
| 7 | Jalisco | Moreno-Letelier et al. [27] |
| 8 | Tamaulipas | McClung de Tapia [123] |
| 9 | Panama | McClung de Tapia [123]; Piperno and Pearsall [40] |
| 10 | Pubenza | van der Hammen and Urrego [129] |
| 11 | Popayán | Aceituno and Loaiza [130] |
| 12 | Santa Elena | Piperno and Stothert [132] |
| 13 | HuacaPrieta | Dillehay et al. [135] |
| 14 | Chilca Valley | Pearsall [134] |
| 15 | Tres Ventanas Cave | Pearsall [134] |
| 16 | Guitarrero Cave | Pearsall [134]; Debouck [126] |
| 17 | Serra da Capivara | Chaves [154]; Chaves and Reinhard [155]; Lahaye et al. [138] |
| 18 | Pedra Pintada | Roosevelt et al. [148]; Roosevelt [139,147]; Shock and Moraes [149] |
| 19 | Peña Roja | Morcote-Ríos et al. [141,142,145] |
| 20 | Cerro Azul | Morcote-Ríos et al. [144] |
| 21 | Carajás | Magalhães [150]; Santos et al. [151] |
| 22 | Llanos de Mojos | Lombardo et al. [152] |
| 23 | Teotônio | Watling et al. [153] |
| 24 | Lapa do Santo | Ortega [156] |
| 25 | Lapa Grande do Taquaraçu | Angeles Flores [157]; Angeles Flores et al. [158] |
| 26 | Santana do Riacho | Resende and Prous [159] |
| 27 | Forte Shell mound | Scheel-Ybert [160]; Scheel-Ybert and Boyadjian [161] |

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
