# Peer review of "Disentangling Domestication from Food Production Systems in the Neotropics"

_quaternary, doi:10.3390/quat4010004_

Round 1
Reviewer 1 Report
This is a detailed and thorough review paper. I do want to highlight that I am not necessarily an expert on plant domestication in the Neotropics. My background focuses mostly on past climate changes, and I have some experience with documenting past human impacts in the tropics. From that perspective, I found the paper insightful and interesting to read. I learned a lot from this paper and I will probably cite it in upcoming works. I think the paper is a good resource for scholars interested in the Neotropics.
I do hope someone with a background in archaeobotany can review this paper because I cannot comment on those details.
One minor comment: Did the authors mean "Altiplano" instead on "plan alto"?
Otherwise, I think the paper is fine as-as.
Author Response
Response: Thank you for catching the Altiplano. We appreciate your time to review and hope you will enjoy the revised manuscript also. Please see our responses to all reviewers in the attached file.

Reviewer 2 Report
This original contribution is a much needed update on the state of our understanding with regard to domestication and food production of the South American neotropics. I recommend accept with minor revisions, below.
This article feels like two papers: the first is a discussion of tempo and mode of domestication that varies according to regional cultures and ecosystems, with supporting evidence and the provocative assertion that so-called centers of agriculture are in fact accumulation points. The second is a scholarly or philosophical treatment that includes a detailed etymology of domestication, a detour into the routes of the first peopling of South America, and other areas that do not contribute to the first premise. As a result, the stoppage of flow detracts from the main impact of the paper.
Strongly recommend that the scholarly definition/history/etymology sections be removed, or reduced and their connection to the main premise clearly described.
In addition, the detailed discussions of regional plant ecosystems are excellent but the lowlands section is really only about Amazonia (the pampa, pantanal, and other important lowland areas are only mentioned), so this section should be titled as lowland Amazonia only.
Minor proofreading needed, e.g. is Soviet capitalized? Some tables jump the page, etc.
Author Response
>This original contribution is a much needed update on the state of our understanding with regard to domestication and food production of the South American neotropics. I recommend accept with minor revisions, below.
Response: Thank you for your careful review.
>This article feels like two papers: the first is a discussion of tempo and mode of domestication that varies according to regional cultures and ecosystems, with supporting evidence and the provocative assertion that so-called centers of agriculture are in fact accumulation points. The second is a scholarly or philosophical treatment that includes a detailed etymology of domestication, a detour into the routes of the first peopling of South America, and other areas that do not contribute to the first premise. As a result, the stoppage of flow detracts from the main impact of the paper.
>Strongly recommend that the scholarly definition/history/etymology sections be removed, or reduced and their connection to the main premise clearly described.
Response: Rather than eliminate one paper, we have attempted to better integrate the parts, after eliminating the detour and a few other paragraphs. Rather than remove the definitions, we expanded them (in the form of an Appendix to avoid overloading the section). As you will see in the Appendix A, current definitions vary according to the interests of the authors. Almost all of them concentrate on one or two human behaviors (selection, cultivation), rather than identifying others that are equally important (care, accumulation, duty). We feel that returning to the dictionary now and then can be constructive. Please see our additional reasoning on lines 234-244.
>In addition, the detailed discussions of regional plant ecosystems are excellent but the lowlands section is really only about Amazonia (the pampa, pantanal, and other important lowland areas are only mentioned), so this section should be titled as lowland Amazonia only.
Response: In section 4.3, we had specifically tried to give more attention to the Atlantic Forest (350 words), the Cerrado (187 words), the Caatinga (122 words), because Amazonia (312 words) is better studied along these lines. We have added additional information about the Cerrado (76 words) and the Caatinga (40 words) to improve the balance. Please see lines 905-906, 919-913 and 926-929, respectively. In other parts of the text, Amazonia does get more attention than the other biomes, but that is because more work has been done there about the topics of interest to this manuscript.
>Minor proofreading needed, e.g. is Soviet capitalized? Some tables jump the page, etc.
Response: Thank you for catching this detail. We have reformatted the table, but will only make it fit the page if the manuscript is accepted.
Please see our responses to all reviewers in the attached file.

Reviewer 3 Report
An excellent, seemingly comprehensive, state-of-the-field review of the archaeobotany of central and south America. Although not my region of interest, there is plenty here that will be of interest to scholars globally.
Author Response
Response: We appreciate your time to review and hope you will enjoy the revised manuscript also. Please see our responses to all reviewers in the attached file.

Reviewer 4 Report
The manuscript entitled ‘Disentangling domestication from food production systems in the Neotropics’ is a relevant contribution about the conceptualization of ‘domestication’, and provides a significant synthetic review of plant use during early and Middle Holocene in Mesoamerica and South America. In that sense, it is remarkable the valuable review effort of information about useful plant species in section. The authors contribute with ethnobotanical data, which is interesting to address the topic of the paper, although more caution should be taken in the use of ethnographic and ethnobotanical knowledge to address past human activities. This paper is interesting as a contribution to better understand plant domestication by providing an alternative model in the Neotropics.
One of the most relevant contributions of this paper is the low dependence of Neotropics populations on agricultural products when compared with Near East and Europe processes of domestication, as well as the persistence of gathering activities and its important role in subsistence in the Neotropics.
In addition, I found very interesting the definition of domesticated landscapes. Although other concepts are more common in the literature, as anthropogenic landscapes and social landscapes, I think that the development and the arguments of the paper support the use of this term, as the described management of landscape lead to the expansion of those species intensively exploited by humans. I recommend the authors to focus the paper on this concept, the landscape domestication by hunter-gatherers. This is a relevant contribution and I recommend removing the confuse ‘food production systems’ concept from the title and add ‘landscape domestication’. Nevertheless, I miss references about landscape evolution based on paleoecological studies in the region to assess the impact of the proposed domesticated landscapes during the Early and Middle Holocene. Authors should add these references.
It is also positive the discussion about the concept of domestication. Domestication consists of the appropriation of reproductive cycles of some plants and animals within mechanisms of social production and reproduction (Revelles, 2017). For that reason, the domestication is a process started when humans manage plants and animals, not when morphological or genetic changes occurred. In this line, this paper shows that many plant species do not show clear morphological changes (incipient domestication) in spite of being managed. What it is important to reflect is the change in human-environment interaction since the start of the domestication process (Wright, 1971). From an archaeological point of view, the most important is the social change involved in these human activities, not the morphological or genetic changes experienced by some animals and plants.
Although I agree in many concepts and find an original contribution, there are some concepts and some aspects of the theoretical approach that require major revision, in some cases showing contradictions:
- Concept of food production system. I think the authors will coincide with me if I say that economic activities developed by hunter-gatherer societies (i.e. selection of plants) are productive activities involving food. The equivalence of agriculture-food production and gathering-not food production is not correct. Obtaining food by hunting and gathering should not be considered non-productive activities, because specific knowledge of the geosystem, the application of several techniques and technologies and an organization of labour and tasks are required and are involved in the global productive and reproductive process of hunter-gatherer societies (Piqué, 1999).
In some cases, authors just need to change concepts. For example, in lines 1053-1054, the affirmation would be more appropriate and of relevant significance if it says ‘primary evidence that domestication of plants does not lead necessarily to reliance on agricultural/farming systems’.
- Authors should talk about plant domestication and not just domestication because animals are not considered in this paper (i.e. title, line 135)
- Line 319 (use of fire), please add references about the use of fire
- I can understand the interest on ethnographic studies about human perception of nature but this is not relevant for the topic of this paper. This paper can be a reference in the conceptualisation of domestication regarding some aspects of the manuscript. But, on the other hand, there are continuous references to human cognitive perception and interaction with plants which are far away from the scientific scope of a journal as Quaternary. For example, in line 139-40, is not necessary for the paper to include the reference of Vrydaghs and Denham suggesting that domestication is a result of human thinking. Another example is all the referent to the agency of plants and animals (line 154, 282). There are some affirmations that are not serious for a scientific paper, as in lines 163, 403 and 434-439. Author must remove these affirmations before considering the publication of this manuscript.
- The authors abuse on the use of Oxford English Dictionary definitions. They should define concepts based on scientific approaches and evidences by experts in the field. These details reduce significantly the level of the paper, which could be really high making some changes. In that sense, removing lines 247-267 would improve the paper.
- I agree that the start of the domestication process was incidental considering that their objective was not domesticating plants, it was just the consumption and use of plants, as stated in lines 270-271. But then, in lines 1054-1058 authors conclude that humans decided what and how domesticate in a conscious process. Firstly, authors had talked about a co-evolutionary process between plants and humans (a coherent approach), combined with references to agency in plants (out of scope of a scientific approach) and finally they talk about human decisions. I suggest the authors be consistent with the message throughout the paper.
- After talking about plants agency, authors write ‘Humans and their management practices are also biotic components of the niche’. Human activities are not determined by natural laws, it is a social component out of the natural system or geosystem. The geosystem is not used, felt or perceived, but exists in a relationship with productive forces and social categories (Beroutchachvili and Bertrand, 1978). Human agency can be disentangled from the biotic component of geosystem, an this is not the case of the proposed plants agency.
- The argumentation of differences between horticulture and agriculture (again based on dictionary definitions) and between Neotropics and Near East is not valid. First agriculture during Neolithic in the Near East and Europe was characterised as intensively managed in small plots, defined also as garden agriculture (Bogaard, 2004; Bogaard and Jones, 2007; Bogaard et al., 2011). I suggest to remove this part of the text and focus the paper on other concepts better developed and more significant for the paper, as the concept of domesticated landscapes.
- In Section 3, authors should mention explicitly for which species there are clear evidence of cultivation and which were just gathered. It would be very appreciated a synthetic table including this information together with site, chronology and references.
- Lines 974-979. It seems that the discussion of this work is mainly based on ethnographic projection from present to past and not base on archaeological evidence. The kind of proposals and hypothesis defended in this paper should have a strong archaeological basis about past plant management practices.
References
Bogaard, A., 2004. Neolithic Farming in Central Europe. Routledge, London.
Bogaard, A., Jones, G., 2007. Neolithic farming in Britain and central Europe: contrast or continuity? In: Whittle, A., Cummings, V. (Eds.), Going Over: the Mesolithic– Neolithic Transition in North-west Europe. British Academy, London, pp. 357–375.
Bogaard, A., Strien, H.-C., Krause, R., 2011. Towards a social geography of cultivation and plant use in an early farming community: Vaihingenan der Enz, south-west Germany. Antiquity 85 (328), 395–416.
Beroutchachvili, N., Bertrand, G., 1978. Le géosystème ou “système territorial naturel”. Revue géographique des Pyrénées et du Sud-Ouest, Toulouse 49 (2), 167–180.
Piqué, R., 1999. Producción y uso del combustible vegetal: una evaluación arqueológica. Treballs d' Etnoarqueologia 3. Universidad Autónoma de Barcelona, CSIC, Madrid.
Revelles, J., 2017. Archaeoecology of Neolithisation. Human-environment interactions in the NE Iberian Peninsula during the Early Neolithic. J. Archaeol. Sci. Rep. 15, 437e445.
Wright, G.A., 1971. Origins of food production in Southwestern Asia: a survey of ideas. Curr. Anthropol. 12 (4–5), 447–475.
Author Response
>The manuscript entitled ‘Disentangling domestication from food production systems in the Neotropics’ is a relevant contribution about the conceptualization of ‘domestication’, and provides a significant synthetic review of plant use during early and Middle Holocene in Mesoamerica and South America. In that sense, it is remarkable the valuable review effort of information about useful plant species in section. The authors contribute with ethnobotanical data, which is interesting to address the topic of the paper, although more caution should be taken in the use of ethnographic and ethnobotanical knowledge to address past human activities. This paper is interesting as a contribution to better understand plant domestication by providing an alternative model in the Neotropics.
Response: Thank you for the careful review and important suggestions to improve the manuscript. With respect to using ethnographic and ethnobotanical knowledge to interpret the past, we feel that this is warranted given the very basic human behaviors involved and number of species that appear in both the archaeobotanical record and modern use and management. We have added a sentence to highlight this overlap (lines 688-690).
>One of the most relevant contributions of this paper is the low dependence of Neotropics populations on agricultural products when compared with Near East and Europe processes of domestication, as well as the persistence of gathering activities and its important role in subsistence in the Neotropics.
Response: We believe that this is partially due to the large areas that were not occupied by states in southern Mesoamerica, the northern Andes and all of lowland South America, as we mention at the end of the manuscript (lines 1100-1106).
>In addition, I found very interesting the definition of domesticated landscapes. Although other concepts are more common in the literature, as anthropogenic landscapes and social landscapes, I think that the development and the arguments of the paper support the use of this term, as the described management of landscape lead to the expansion of those species intensively exploited by humans. I recommend the authors to focus the paper on this concept, the landscape domestication by hunter-gatherers. This is a relevant contribution and I recommend removing the confusing ‘food production systems’ concept from the title and add ‘landscape domestication’. Nevertheless, I miss references about landscape evolution based on paleoecological studies in the region to assess the impact of the proposed domesticated landscapes during the Early and Middle Holocene. Authors should add these references.
Response: As you point out, there are numerous terms for this concept. Bruce Smith (Phil. Trans. R. Soc. B (2011) 366, 836–848) provides a table full of synonyms. This concept is gradually being subsumed within cultural niche construction, although we prefer to maintain it. However, it is only a part of what we are trying to show. We also adopted ‘food production systems’ for a reason: to differentiate the Neotropics from the Paleotropics, a point that you raise below also. We purposefully did not delve into the large and expanding literature of the paleoecology of human landscape domestication in the Neotropics precisely because it is so large and expanding so rapidly. As you point out below, this is already a large manuscript.
>It is also positive the discussion about the concept of domestication. Domestication consists of the appropriation of reproductive cycles of some plants and animals within mechanisms of social production and reproduction (Revelles, 2017). For that reason, the domestication is a process started when humans manage plants and animals, not when morphological or genetic changes occurred. In this line, this paper shows that many plant species do not show clear morphological changes (incipient domestication) in spite of being managed. What it is important to reflect is the change in human-environment interaction since the start of the domestication process (Wright, 1971). From an archaeological point of view, the most important is the social change involved in these human activities, not the morphological or genetic changes experienced by some animals and plants.
Response: We agree completely. This is why we spent so much time (and abused the Oxford English Dictionary definitions) to identify additional human behaviors beyond selection and cultivation. Caring for plants (and animals), accumulating their variety, and even feeling a sense of duty to care for them should be part of the definitions. See lines 283-285.
>Although I agree in many concepts and find an original contribution, there are some concepts and some aspects of the theoretical approach that require major revision, in some cases showing contradictions:
>- Concept of food production system. I think the authors will coincide with me if I say that economic activities developed by hunter-gatherer societies (i.e., selection of plants) are productive activities involving food. The equivalence of agriculture-food production and gathering-not food production is not correct. Obtaining food by hunting and gathering should not be considered non-productive activities, because specific knowledge of the geosystem, the application of several techniques and technologies and an organization of labour and tasks are required and are involved in the global productive and reproductive process of hunter-gatherer societies (Piqué, 1999).
>In some cases, authors just need to change concepts. For example, in lines 1053-1054, the affirmation would be more appropriate and of relevant significance if it says ‘primary evidence that domestication of plants does not lead necessarily to reliance on agricultural/farming systems’.
Response: We agree with you that hunter-gatherers have some activities that are classified as food production, in contrast to other activities that can be considered food procurement. This is the reason that we introduced both the “pre-domestication cultivation” and the “low-level food production” terminologies. Essentially, there are no “pure” hunter-gatherer societies in regions where plants are parts of subsistence. One of the ideas that we try to develop is that agriculture, in the original sense of the word (more abuse of Oxford English Dictionary) was not present in the Neotropics before European conquest. In order to make this clear, we adopted the ‘confusing’ term of food production systems.
>- Authors should talk about plant domestication and not just domestication because animals are not considered in this paper (i.e., title, line 135).
Response: Quite correct. We did not change the title, but we did verify that in most instances we emphasized plant domestication.
>- Line 319 (use of fire), please add references about the use of fire.
Response: Added. Please see line 338.
>- I can understand the interest on ethnographic studies about human perception of nature but this is not relevant for the topic of this paper. This paper can be a reference in the conceptualisation of domestication regarding some aspects of the manuscript. But, on the other hand, there are continuous references to human cognitive perception and interaction with plants which are far away from the scientific scope of a journal such as Quaternary. For example, in line 139-40, it is not necessary for the paper to include the reference of Vrydaghs and Denham suggesting that domestication is a result of human thinking. Another example is all the reference to the agency of plants and animals (line 154, 282). There are some affirmations that are not serious for a scientific paper, as in lines 163, 403 and 434-439. Author must remove these affirmations before considering the publication of this manuscript.
Response: Vrydaghs & Denham were removed. However, while some readers of Quaternary may be unfamiliar with plant agency, this is an expanding topic in domestication studies, especially from an anthropological perspective. The references we cite for original line 154 are both by archaeologists. The citation for original line 163 is by an anthropologist and an archaeologist. Original lines 434-439 are thoroughly cited with work by anthropologists and archaeologists. This inclusion of plant agency is meant to highlight that care, an important part of the definition in the Oxford English Dictionary, is important in the social lives of humans that interact with plants, sometimes resulting in domestication. We have added a paper by Christine Hastorf (1998, Antiquity), another archaeologist, who expands on the importance of this when considering domestication. Above, you comment that “it is important to reflect the change in human-environment interaction since the start of the domestication process”. We feel that the care given to accumulated plants is part of this and that this care is both intentional (part of human behavior) and is stimulated by plant response (agency).
>- The authors abuse on the use of Oxford English Dictionary definitions. They should define concepts based on scientific approaches and evidences by experts in the field. These details reduce significantly the level of the paper, which could be really high making some changes. In that sense, removing lines 247-267 would improve the paper.
Response: Rather than remove lines 247-267, we have added a number of current “scientific” definitions in the form of Appendix A. As you will see, these “standard” definitions concentrate on only two behaviors: selection and cultivation. They vary according to the objectives of the authors of each paper. We feel that returning to a dictionary can help tease out more behaviors and we feel that we have been rewarded. Care, accumulation and duty should be important parts of any definition of domestication. Care, because it starts even before humans accumulate, as when they protect plants in ecosystems, resulting in incidental domestication and landscape domestication. Once they accumulate plants in and around the settlement, care becomes even more important. Accumulation is essential. In fact, one of us (CRC) thinks that this is more important than selection. The reason: this accumulation is responsible for “The variation of animals and plants under domestication”, the title of Darwin’s magnum opus. Without this accumulation, the amount of variation available for selection is limited to that in each natural population. With this accumulation, not only is there more variation available for selection, but the variation increases naturally via hybridization in and around human settlements. Duty is important because it guarantees care, which is essential for any food production system.
>- I agree that the start of the domestication process was incidental considering that their objective was not domesticating plants, it was just the consumption and use of plants, as stated in lines 270-271. But then, in lines 1054-1058 authors conclude that humans decided what and how domesticate in a conscious process. Firstly, authors had talked about a co-evolutionary process between plants and humans (a coherent approach), combined with references to agency in plants (out of scope of a scientific approach) and finally they talk about human decisions. I suggest the authors be consistent with the message throughout the paper.
Response: Thank you for catching our lapse in lines 1054-1058. This has now been changed (lines 1102-1103).
>- After talking about plants agency, authors write ‘Humans and their management practices are also biotic components of the niche’. Human activities are not determined by natural laws, it is a social component out of the natural system or geosystem. The geosystem is not used, felt or perceived, but exists in a relationship with productive forces and social categories (Beroutchachvili and Bertrand, 1978). Human agency can be disentangled from the biotic component of geosystem, and this is not the case of the proposed plants agency.
Response: We agree that plant agency cannot be disentangled from the biotic components of the niche. However, when humans act to change biotic and abiotic factors in the niche, such as irrigate, fertilize, weed, plant agency can be disentangled, because plants respond to these changes (see lines 398-403 in original manuscript). In lines 418-420, we added the social aspect.
>- The argumentation of differences between horticulture and agriculture (again based on dictionary definitions) and between Neotropics and Near East is not valid. First agriculture during Neolithic in the Near East and Europe was characterized as intensively managed in small plots, defined also as garden agriculture (Bogaard, 2004; Bogaard and Jones, 2007; Bogaard et al., 2011). I suggest to remove this part of the text and focus the paper on other concepts better developed and more significant for the paper, as the concept of domesticated landscapes.
Response: You are quite correct. We have added a phrase to correct this misconception. Lines 444-446. Thank you for the citations, but because they are focused more on Europe, we used others focused on SW Asia.
>- In Section 3, authors should mention explicitly for which species there are clear evidence of cultivation and which were just gathered. It would be very appreciated a synthetic table including this information together with site, chronology and references.
Response: We did not accept this suggestion, since it would involve creating a table with nearly 13,000 lines (the sums of Table 1). The current curators of the 3 databases we use are coauthors and have agreed to make information available to interested researchers.
>- Lines 974-979. It seems that the discussion of this work is mainly based on ethnographic projection from present to past and is not based on archaeological evidence. The kind of proposals and hypothesis defended in this paper should have a strong archaeological basis about past plant management practices.
Response: Earlier in our response, we justified ethnographic projection, based on the very basic human behaviors involved and the fact that many of the species in the archaobotanical record are still used, managed and often domesticated. We added some justification for our projection. Please see lines 688-690, 773-776.
Please see our responses to all reviewers in the attached file.

Round 2
Reviewer 4 Report
Dear,
The authors have improved the manuscript following some reviewers' suggestions. In other cases, they justified why they did not follow the suggestions in their response to comments. The paper is a relevant contribution and I find that the different theoretical statements are properly argued. I enjoyed and learnt reading this manuscript and recommend its publication.
best regards